# Massive colonization of protein-coding exons by selfish genetic elements in *Paramecium* germline genomes

Diamantis Sellis[1☯¤a], Frédéric Guérin[2☯¤b], Olivier Arnaiz[3], Walker Pett[1], Emmanuelle Lerat[1], Nicole Boggetto[2], Sascha Krenek[4], Thomas Berendonk[4], Arnaud Couloux[5], Jean-Marc Aury[5], Karine Labadie[6], Sophie Malinsky[7,8], Simran Bhullar[7], Eric Meyer[7], Linda Sperling[3], Laurent Duret[1‡*], Sandra Duharcourt[2‡*]

**1** Université de Lyon, CNRS, Laboratoire de Biométrie et Biologie Evolutive UMR 5558, Villeurbanne, France, **2** Université de Paris, CNRS, Institut Jacques Monod, Paris, France, **3** Université Paris-Saclay, CEA, CNRS, Institute for Integrative Biology of the Cell (I2BC), Gif-sur-Yvette, France, **4** TU Dresden, Institute of Hydrobiology, Dresden, Germany, **5** Génomique Métabolique, Genoscope, Institut de biologie François Jacob, CEA, CNRS, Université d'Évry, Université Paris-Saclay, Evry, France, **6** Genoscope, Institut de biologie François-Jacob, Commissariat à l'Energie Atomique (CEA), Université Paris-Saclay, Evry, France, **7** Institut de Biologie de l'Ecole Normale Supérieure (IBENS), Ecole Normale Supérieure, CNRS, INSERM, Université PSL, Paris, France, **8** Université de Paris, Paris, France

☯ These authors contributed equally to this work.
¤a Current address: CosmoTech, 5 Passage du Vercors, Lyon, France
¤b Current address: Scipio bioscience, Montrouge, France
‡ LD and SD also contributed equally to this work.
* Laurent.Duret@univ-lyon1.fr (LD); sandra.duharcourt@ijm.fr (SD)

**Data Availability Statement:** All relevant data are within the paper and its Supporting Information files. All sequences and genome assemblies have been deposited in the public nucleotide archive

## Abstract

Ciliates are unicellular eukaryotes with both a germline genome and a somatic genome in the same cytoplasm. The somatic macronucleus (MAC), responsible for gene expression, is not sexually transmitted but develops from a copy of the germline micronucleus (MIC) at each sexual generation. In the MIC genome of *Paramecium tetraurelia*, genes are interrupted by tens of thousands of unique intervening sequences called internal eliminated sequences (IESs), which have to be precisely excised during the development of the new MAC to restore functional genes. To understand the evolutionary origin of this peculiar genomic architecture, we sequenced the MIC genomes of 9 *Paramecium* species (from approximately 100 Mb in *Paramecium aurelia* species to >1.5 Gb in *Paramecium caudatum*). We detected several waves of IES gains, both in ancestral and in more recent lineages. While the vast majority of IESs are single copy in present-day genomes, we identified several families of mobile IESs, including nonautonomous elements acquired via horizontal transfer, which generated tens to thousands of new copies. These observations provide the first direct evidence that transposable elements can account for the massive proliferation of IESs in *Paramecium*. The comparison of IESs of different evolutionary ages indicates that, over time, IESs shorten and diverge rapidly in sequence while they acquire features that allow them to be more efficiently excised. We nevertheless identified rare cases of IESs that are under strong purifying selection across the *aurelia* clade. The cases examined contain or overlap cellular genes that are inactivated by excision during development, suggesting

(accession numbers: PRJEB40886 and PRJEB42373). All detected IESs and their annotation have been deposited at https://doi.org/10.5281/zenodo.4836464. This archive also contains the list of mobile IESs and their alignments, the list of highly conserved IESs and their alignments. All scripts used in the analysis are available at https://github.com/sellisd/IES All numerical values underlying the Figures may be found at https://doi.org/10.5281/zenodo.4836464.

**Funding:** This work was supported by the Centre National de la Recherche Scientifique (https://cnrs.fr), by the Agence Nationale de la Recherche (https://anr.fr) (ANR-18-CE12-0005 to EM, LD, SD; ANR-19-CE12-0015 to SD, OA). It received support under the program "Investissements d'Avenir" launched by the French Government and implemented by ANR with the references ANR-10-LABX-54 MEMOLIFE and ANR-10-IDEX-0001-02 PSL Research to EM. It was supported by the Fondation de la Recherche Medicale (https://don.frm.org)(Equipe FRM DEQ20160334868) to SD and by Labex Who Am I? (http://www.labex-whoami.org/fr)(ANR-11-LABX-0071) "Initiatives d'excellence" (Idex ANR-11-IDEX-0005-02) to SD. The sequencing effort was funded by France Génomique (https://www.france-genomique.org) through involvement of the technical facilities of Genoscope (ANR-10-INBS-09-08) to SD. We acknowledge the ImagoSeine facility, member of the France BioImaging infrastructure supported by the ANR-10-INSB-04. DS and FG received a salary from Agence Nationale de la Recherche. The funders had no role in study design, data collection and analysis, decision to publish, or preparation of the manuscript.

**Competing interests:** The authors have declared that no competing interests exist.

**Abbreviations:** GFP, green fluorescent protein; IES, internal eliminated sequence; IRS, IES retention score; ITm, IS630-Tc1-mariner; MAC, macronucleus; MIC, micronucleus; ORF, open reading frame; Pgm, PiggyMac; TE, transposable element; TIR, terminal inverted repeat; WGP, wheat grass powder.

conserved regulatory mechanisms. Similar to the evolution of introns in eukaryotes, the evolution of *Paramecium* IESs highlights the major role played by selfish genetic elements in shaping the complexity of genome architecture and gene expression.

## Introduction

In multicellular organisms, the division of labor between transmission and expression of the genome is achieved by separation of germline and somatic cells. Such a division is also observed in some unicellular eukaryotes, including ciliates [1]. The ciliate *Paramecium tetraurelia* separates germline and somatic functions into distinct nuclei in the same cell. Somatic functions are supported by the highly polyploid macronucleus (MAC) that is streamlined for gene expression and destroyed at each sexual cycle. Germline functions are ensured by 2 small, diploid micronuclei (MIC) that are transcriptionally silent during vegetative growth. During sexual events, the MICs undergo meiosis and transmit the germline genome to the zygotic nucleus. New MICs and new MACs differentiate from mitotic copies of the zygotic nucleus. MAC differentiation involves massive and reproducible DNA elimination events (for review: [2,3]). In addition to the variable elimination of large regions containing repeats, approximately 45,000 unique, short, interspersed internal eliminated sequences (IESs) are precisely removed from intergenic and coding regions [4,5]. Precise excision of IESs at the nucleotide level is essential to restore functional cellular genes, since 80% of the IESs are inserted within protein-coding genes, and about half of the approximately 40,000 genes are interrupted by IESs. IESs are invariably bounded by two 5′-TA-3′ dinucleotides, one of which is left at the junction in the MAC genome after excision. IES excision in the developing MAC is initiated by DNA double-strand breaks at IES ends by the endonuclease PiggyMac (Pgm) assisted by other proteins, which are likely part of the excision machinery or interact with it [6–9].

Despite significant progress in characterization of the mechanisms underlying IES elimination, the evolutionary origin of IESs remains mysterious. On the basis of sequence similarities between the consensus found adjacent to the TA dinucleotide at IES ends and the extremities of DNA transposons from the IS630-Tc1-mariner (ITm) superfamily, Klobutcher and Herrick hypothesized that IESs might be degenerated remnants of transposable elements (TEs) [10,11]. This hypothesis was further substantiated by the discovery that the endonuclease responsible for IES excision in *P. tetraurelia* is encoded by a domesticated PiggyBac transposase [6], assisted by a related family of catalytically inactive transposases [7]. All-by-all sequence comparison of the *P. tetraurelia* 45,000 IESs and of their flanking sequences identified 8 families of "mobile IESs" (2 to 6 copies), i.e., homologous IESs inserted at nonhomologous sites in the genome [4]. One such family (with 6 copies) was found similar to the terminal inverted repeats (TIRs) of *Thon*, a DNA transposon of the ITm superfamily, indicating that some IESs derive from TEs [4]. These cases provided support to the notion that at least some IESs have derived from recently mobilized elements. However, the rather small number of mobile IESs detected (23 copies out of 45,000 IESs) suggested a limited activity of transposable IESs in the recent evolutionary history of the *P. tetraurelia* lineage [4]. There is also evidence that some IESs originated from MAC sequences, as described, for example, for the IESs involved in mating type determination in several species [12,13]. The extent to which the 45,000 IESs detected in *P. tetraurelia* derive from TEs or from MAC sequences therefore remained unclear.

In order to gain insight into the evolutionary origin of IESs in the *Paramecium* lineage, we adopted a comparative genomic approach. *P. tetraurelia* belongs to the *Paramecium aurelia*

group of species that comprises a score of morphologically similar yet genetically isolated species [14–17] (see [13] for a detailed description of the species). Here, we sequenced the germline MIC genomes of 8 *P. aurelia* species and 1 outgroup (*Paramecium caudatum*), compared the IES repertoire across these 9 species, and analyzed the evolutionary trajectories of IESs in the *Paramecium* lineage.

## Results

### Sequencing of somatic and germline genomes in 9 *Paramecium* species: Gigantic germline genome in *P. caudatum*

In order to determine the origin and evolution of IESs in the *Paramecium* lineage, we sequenced the germline MIC genome and the somatic MAC genome of several *Paramecium* species. We selected 8 species from the *P. aurelia* complex and 1 outgroup species, *P. caudatum*, which diverged from the *aurelia* complex before the 2 most recent *Paramecium* whole-genome duplications [15]. To sequence the germline MIC genome, we purified the germline nuclei (MICs) of each species using a flow cytometry procedure that we previously developed for *P. tetraurelia* [5] (S1 Fig) and used paired-end Illumina sequencing (see Materials and methods and S1 Table).

To estimate the size of the MIC genomes, we employed 2 distinct approaches. First, we used the MIC-enriched preparations from *Paramecium* cultures to yield values for DNA quantity in the MICs by flow cytometry (see Materials and methods and S1 Data). The estimated MIC genome sizes are within a similar range (140 to 173 Mb) for the *P. aurelia* species, except for *P. sonneborni* (S1 Table), which was estimated to be roughly the double of the others. The second, independent approach for genome size estimations was based on the sequence reads themselves and used the k-mer method described in [18,19]. The estimated MIC genome sizes were comprised between 108 Mb to 123 Mb for the *aurelia* species, with a considerably larger MIC genome (283 Mb) for *P. sonneborni* (Table 1). While the values obtained using the flow cytometry method were greater than those with the k-mer method, the estimated MIC genome sizes were within a similar range for both methods (S3 Fig).

With both methods, the estimated MIC genome size of *P. caudatum* strain My43c3d was the largest among the species analyzed (approximately 1,300 to 1,600 Mb). To confirm this observation, we estimated the genome size of 9 additional strains belonging to the 2 major clades A and B described in the *caudatum* lineage, as well as another divergent strain [20]. The data confirmed that the MIC genome size in the *caudatum* lineage is far bigger than that in the *aurelia* lineage and revealed great variations of genome size among the different strains (from 1,600 Mb to 5,500 Mb), even within the same clade (S1 Data). To investigate the composition of the gigantic *caudatum* genomes, we searched for the presence of repeats in the MIC sequence reads of strain My43c3d. We identified 2 major satellite repeats, Sat1 and Sat2 (332 bp and 449 bp, respectively), which represent 42% and 29%, respectively, of the MIC genome (S4 Fig). Both Sat1 and Sat2 repeats were detected in the *P. caudatum* strains of the clade B, to which the strain My43c3d belongs, but not in the other *P. caudatum* strains (S4 Fig). Thus, the repeats are not shared by all *P. caudatum* strains and most likely invaded the MIC genome after the divergence between clades A and B.

The MAC genome was sequenced for 4 *aurelia* species that had not been sequenced previously (S2 Table). We defined the "constitutive" MAC genome as the DNA sequences retained in all MAC copies (see Materials and methods and S2 Table). The size of the constitutive MAC genome assembly was similar among *P. aurelia* species (66 to 73 Mb) with a noticeably larger size for *P. sonneborni* (83 Mb) (Table 1). The number of protein coding genes follows a similar distribution (36,179 to 42,619) in *aurelia* species, with a larger number of genes in *P.*

**Table 1. Characteristics of analyzed genomes.** Species are ordered according to the phylogeny (Fig 1). (a) MIC genome size (in Mb) was estimated based on k-mer counts (see S1 Table for additional estimates based on flow cytometry). (b) Sequencing depth in MIC-specific regions estimated from the distribution of k-mer counts (S2 Fig). (c) Size of constitutive MAC genome assembly. (d) The sensitivity of IES detection was limited in *P. caudatum* and *P. sonneborni* because of the relatively low sequencing depth of their MIC genome. Estimates of total number of IESs, based the IES density observed in regions with sufficient read depth (see e), are indicated in parenthesis for these 2 species. (e) IES density measured in MAC-destined sequences, after exclusion of regions with insufficient MIC read depth (<15X).

| Species (strain) | MIC genome | | MAC-destined regions | | | |
|---|---|---|---|---|---|---|
| | Size (Mb) (a) | Sequencing depth (b) | Size (Mb) (c) | Nb. of protein genes | Nb. of IESs (d) | IES density (per kb) (e) |
| *P. tetraurelia* (51) | 108 | 31 | 72 | 40,460 | 44,128 | 0.62 |
| *P. octaurelia* (138) | 108 | 56 | 72 | 44,398 | 44,509 | 0.61 |
| *P. biaurelia* (V1-4) | 119 | 58 | 74 | 40,261 | 45,384 | 0.65 |
| *P. tredecaurelia* (209) | 127 | 71 | 65 | 36,179 | 42,275 | 0.66 |
| *P. pentaurelia* (87) | 112 | 194 | 72 | 41,676 | 42,686 | 0.57 |
| *P. primaurelia* (AZ9-3) | 114 | 65 | 73 | 42,619 | 43,766 | 0.59 |
| *P. sonneborni* (ATCC 30995) | 286 | 13 | 82 | 49,951 | 60,198 (approximately 85,000) | 1.05 |
| *P. sexaurelia* (AZ8-4) | 123 | 141 | 68 | 36,094 | 47,002 | 0.70 |
| *P. caudatum (My43c3d)* | 1,300 | 13 | 30 | 18,673 | 8,762 (approximately 15,000) | 0.47 |

IES, internal eliminated sequences; MAC, macronucleus; MIC, micronucleus.

*sonneborni* (Table 1). This contrasts with the much smaller MAC genome size and number of genes of the outgroup *P. caudatum* [15].

In conclusion, the 8 species of the *aurelia* complex that we analyzed share similar genome characteristics, with an MIC genome of approximately 110 to 160 Mb, 50% to 70% of which is retained during MAC development (approximately 70 to 80 Mb). The only notable exception is *P. sonneborni*, with a 300 to 400 Mb MIC genome, of which about 25% is retained in its MAC. The MIC genome of the outgroup *P. caudatum* is much larger (approximately 1,300 to 1,600 Mb). Only 2% of MIC sequences are retained in the MAC of *P. caudatum* strain My43c3d, and 83% of the MIC-specific sequences consist of repeated DNA (S4 Fig).

## IES repertoire

IESs were identified by comparing MIC sequence reads to the MAC genome assembly (see Materials and methods; [4,21]). Overall, the number of detected IESs in MAC-destined sequences is similar across *Paramecium* species (approximately 42,000 to 47,000 IESs), with the exception of *P. sonneborni* (approximately 60,000 IESs) and *P. caudatum* (approximately 9,000 IESs). It should be noted that the sensitivity of IES detection depends on sequencing depth (S5 Fig). For most species, we expect that >99% of IESs have been detected, except in *P. caudatum*, and *P. sonneborni*, where the sequencing depth was limited, due to the unexpected large size of their MIC genome (Table 1). To circumvent this issue, we compared the IES density across species by taking into account only IESs annotated in regions with at least 15X depth of MIC sequence reads mapped onto the MAC assembled genome (Table 1): In *P. caudatum*, the density of detected IES sites in MAC-destined regions (0.5 IESs per kb) is only slightly lower than in other species (approximately 0.6 IESs per kb). This suggests that the genome of *P. caudatum* probably contains about 15,000 IESs in its MAC-destined regions.

Our approach is designed to identify IESs only if they are present within loci retained in the MAC. Hence, IESs located in MIC-specific regions (e.g., IESs nested within other IESs [22,23]) remain undetected. Interestingly, in 4 species whose MAC genome was sequenced at very high depth *(Paramecium octaurelia, Paramecium primaurelia, Paramecium pentaurelia,* and *P. sonneborni)*, the initial MAC genome assemblies included 10 to 16 Mb of MAC-variable regions, which correspond to DNA sequences that are not completely eliminated and, instead, are

retained in a small fraction of MAC copies (see Materials and methods, S6 Fig and S2 Table). We identified many IESs in these regions, at a density (0.4 to 0.5 IESs per kb, S2 Table) nearly as high as in MAC-destined regions (Table 1). This suggests that in addition to IESs located in MAC-destined regions, many other IESs are present within MIC-specific regions.

In all species, the vast majority of IESs in MAC-destined regions (73% to 81%) are located in protein-coding exons, and approximately 5% are located in introns. Overall, there is a slight enrichment of IESs within genes (S3 Table). This enrichment is not true for all gene categories. In particular, we observed a depletion of IESs in highly expressed genes: On average, the IES density in the top 10% most expressed genes is 37% lower than in the bottom 10% (S7 Fig). This pattern, consistent with previous observations in *P. tetraurelia*, suggests that IES insertions are counterselected in highly expressed genes [4].

## Age distribution of IESs

In order to explore the origin and evolution of IESs, we resolved the phylogenetic relationship among the sequenced species. To do so, we classified all protein sequences into families ($N$ = 13,617 gene families) and inferred the species phylogeny using the subset of 1,061 gene families containing 1 single sequence from each species. In agreement with previous reports [16,24], we found strong support for a division of the *aurelia* complex in 2 subclades (hereafter referred to as subclades A and B), separating *P. sonneborni* and *Paramecium sexaurelia* from the other *aurelia* species (Fig 1). We then used this species phylogeny to identify gene duplications and speciation events in each of the 13,617 gene families, using the PHYLDOG tree reconciliation method [25].

In order to date events of IES gain or loss, we mapped the position of IES excision sites in multiple alignments of each gene family (nucleic sequence alignments based on protein alignments): IESs located at the exact same position within a codon were assumed to derive from a single ancestral insertion event, whatever their present level of sequence similarity (S9 Fig). IESs located at homologous sites are hereafter referred to as "co-orthologous" IESs. We use the term "co-orthologous" rather than "orthologous," because these sets frequently include paralogs, notably as a result of whole-genome duplications. To avoid ambiguities due to low-quality alignments, we only analyzed IESs present within well-conserved protein-coding regions (which represent from 45% to 51% of IESs located in coding regions; Fig 1). We then used the reconciled gene tree to map events on the species phylogeny and estimate rates of IES gain and loss along each branch of the species tree using a Bayesian approach accounting for IES losses and missing data (see Materials and methods). In the absence of fossil records, it is impossible to date speciation events (in million years). We therefore used sequence divergence (number of amino acid substitutions per site) along branches of the phylogeny as a proxy for time.

Overall, 10.8% of IESs detected in *aurelia* species predate the divergence from *P. caudatum* (referred to as "Old" IESs in Fig 1), 79% were gained after the divergence of *P. caudatum*, but before the radiation of the *aurelia* complex ("Intermediate" in Fig 1) and 10.2% are more recent. The rate of IES gain varied widely over time: A burst of insertions occurred in the ancestral branch leading to the *aurelia* clade, followed by a progressive slowdown in most lineages, except in *P. sonneborni* where the rate of IES gain strongly increased again in the recent period (18.8% of IESs detected in *P. sonneborni* are specific to that species). The IES gain rate has remained substantial in *P. sexaurelia* and *Paramecium tredecaurelia* but has dropped to very low levels in *P. tetraurelia/P.octaurelia* and in *P. pentaurelia/P. primaurelia* lineages, about 20 times lower than in *P. sonneborni* or in the ancestral *aurelia* lineage (Fig 1). The rate of IES loss appears to be more uniform along the phylogeny, with only 2- to 3-fold variation (Fig 1).

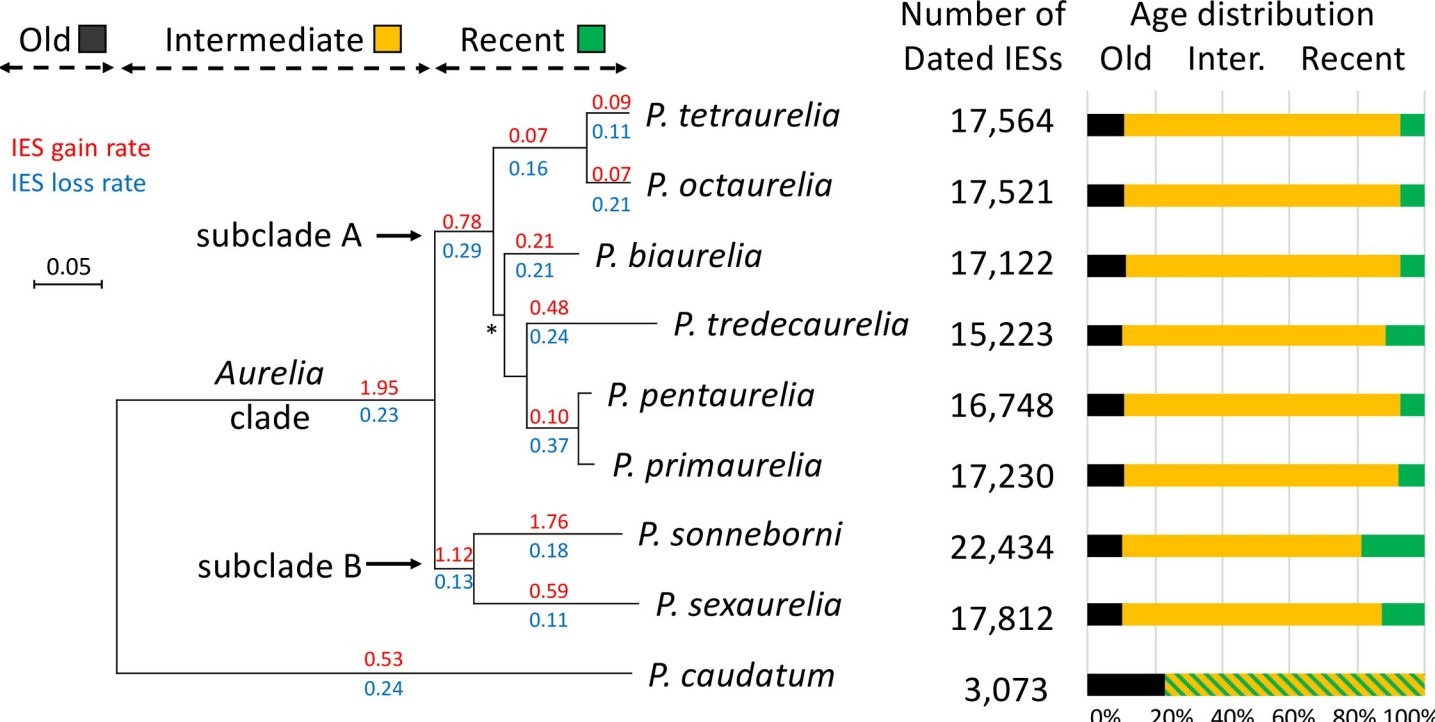

**Fig 1. Dynamics of IES insertion/loss in *Paramecium*.** The species phylogeny was reconstructed from a concatenated alignment of 1,061 single-copy genes. All internal branches are supported by 100% bootstrap values (except branch *: bootstrap support = 83%). The age of IESs located within coding regions was inferred from the pattern of presence/absence within gene family alignments (*N* = 13,617 gene families). Only IESs present within well-aligned regions were included in this analysis. The number of dated IESs and the fraction predicted to be old (predating the divergence between *P. caudatum* and the *P. aurelia* lineages), intermediate (before the radiation of the *P. aurelia* complex), or recent are reported for each species. Rates of IES gain (in red) and loss (in blue) were estimated along each branch using a Bayesian approach. Gain rates are expressed per kb per unit of time (using the branch length—in substitutions per site—as a proxy for time). Loss rates are expressed per IES per unit of time (see S8 Fig for more details on the measure of gain/loss rates). NB: Estimates of loss rate along terminal branches of the phylogeny also include false negatives (i.e., IESs that are present but that have not been detected) and hence may be overestimated. The data underlying this figure may be found at https://doi.org/10.5281/zenodo.4836464. IES, internal eliminated sequence.

## Recent waves of mobilization of IESs

The episodic bursts of IES gains that we observed in the phylogeny are reminiscent of the dynamics of invasion by TEs. To test the hypothesis that IESs might correspond to TEs, we searched for evidence of mobile IESs, i.e., IES sequences sharing significant sequence similarity but inserted at different (nonhomologous) loci. In a first step, we compared all IESs against each other with BLASTN to identify clusters of similar IESs. Consistent with previous observations made in *P. tetraurelia* [4], we found that the vast majority of IESs correspond to unique sequences in all MIC genomes, but a fraction of IESs are present in multiple copies (Fig 2A). In a second step, all clusters with ≥10 interspersed copies were manually inspected, to precisely delineate the boundaries of the repeated element and create a multiple alignment of full-length copies. We then used these representative multiple alignments to perform an exhaustive sequence similarity search based on HMM profiles over the entire IES dataset (see Materials and methods). Among the hits, we distinguished 2 categories: (1) cases where the detected copy is located within the IES but does not include the extremities of the IES; and (2) cases where the extremities of the copy correspond precisely to the extremities of the IES. The first category probably corresponds to TEs that were inserted within a preexisting IES (i.e., nested repeats). The second category corresponds to cases where the transposed element is the IES

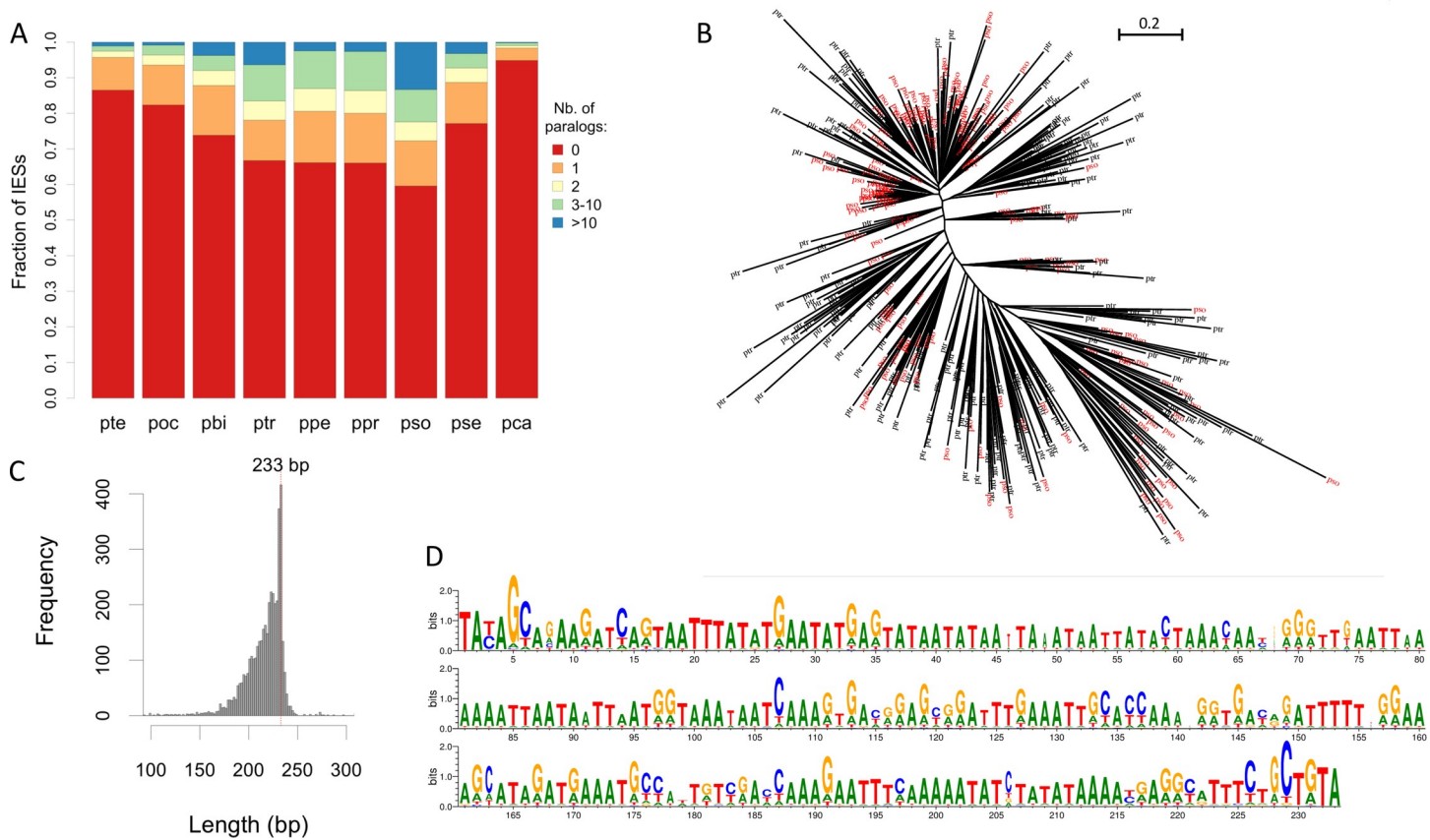

**Fig 2. Proportion of IESs corresponding to repeated sequences and phylogenetic analysis of the largest family of mobile IESs.** (A) The vast majority of IESs correspond to unique sequences, but a fraction of IESs are present in multicopies. For each species, all IESs were compared against each other with BLASTN (with an E-value threshold of $10^{-5}$). The distribution of the number of BLAST hits per IES (excluding self-hits) is displayed for each species. (B) The largest family of multicopy IESs (FAM_2183) includes 4,153 copies. We present here a phylogenetic tree of a subset of sequences (200 IESs from *P. tredecaurelia* in black and 200 from *P. sonneborni* in red), randomly sampled from the entire FAM_2183 alignment (computed with PhyML [27]). The tree topology is mainly star-like, which indicates that most copies derive from several bursts of insertions. (C) Length distribution of the 4,153 FAM_2183 mobile IESs. (D) Sequence logo [28], based on the alignment of the entire FAM_2183 family. The information content reported in the logo was computed relative to the base composition of IESs, to take into account their high AT content. All copies present a high level of sequence similarity (average pairwise identity 72%) throughout their entire length, not just at their ends. The data underlying this figure may be found at https://doi.org/10.5281/zenodo.4836464. IES, internal eliminated sequence.

itself (i.e., mobile IESs). Overall, we detected 24 families with at least 10 copies of mobile IESs, totaling 7,443 copies of mobile IESs (Table 2).

Four of these mobile IESs present homology with DNA transposons of the ITm superfamily previously identified in *P. tetraurelia* [4,5] (Table 2). FAM_2314 (3.4 kb) includes an intact open reading frame (ORF) encoding a DDE transposase. FAM_1294 (1.7 kb) is homologous to *Baudroie*, a composite Tc1-mariner element and includes an ORF with similarity to tyrosine-type recombinases. FAM_1402 (0.7 kb) and FAM_1257 (0.5 kb) correspond to nonautonomous elements, homologous to the TIRs of *Thon* and *Merou*, respectively. The other families of mobile IESs do not match with any known TEs. Their relatively short lengths (32 bp to 765 bp) and the absence of similarity with any known protein indicate that they most probably correspond to nonautonomous elements, mobilized by transposases expressed from active TEs.

The genomic distribution of mobile IESs within MAC-destined regions is similar to that of other IESs: Most of the families are predominantly located within protein-coding regions (which represent approximately 70% of the MAC genome) (S4 Table). The only notable

**Table 2. Taxonomic distribution of mobile IESs.** Copies of mobile IESs were searched among IES sequences (see Materials and methods). Detected copies were divided in 2 categories: nested copies (i.e., copies inserted within an IES, but not including the extremities of the IES) and bona fide mobile IESs (i.e., copies whose extremities correspond to the extremities of the IES). This table lists all families for which at least 1 species contains ≥10 copies of bona fide mobile IESs in its genome (see S4 Table for information on nested copies). Species codes: pbi, *P. biaurelia*; pca, *P. caudatum*; poc, *P. octaurelia*; ppe, *P. pentaurelia*; ppr, *P. primaurelia*; pso, *P. sonneborni*; pse, *P. sexaurelia*; pte, *P. tetraurelia*; ptr, *P. tredecaurelia*.

| Repeat family | Length (bp) | Nb. of mobile IESs | Number of mobile IESs per species | | | | | | | | |
|---|---|---|---|---|---|---|---|---|---|---|---|
| | | | pca | pse | pso | ptr | ppe | ppr | pbi | poc | pte |
| FAM_2183 | 233 | 4,153 | 0 | 4 | 3252 | 897 | 0 | 0 | 0 | 0 | 0 |
| FAM_3 | 290 | 1,783 | 0 | 0 | 0 | 15 | 344 | 321 | 766 | 146 | 191 |
| FAM_2938 | 765 | 378 | 0 | 7 | 370 | 1 | 0 | 0 | 0 | 0 | 0 |
| FAM_2317 | 768 | 331 | 0 | 82 | 140 | 109 | 0 | 0 | 0 | 0 | 0 |
| FAM_2942 | 211 | 110 | 0 | 0 | 110 | 0 | 0 | 0 | 0 | 0 | 0 |
| FAM_2334 | 214 | 106 | 0 | 17 | 89 | 0 | 0 | 0 | 0 | 0 | 0 |
| FAM_2321 | 471 | 84 | 1 | 11 | 58 | 10 | 1 | 1 | 2 | 0 | 0 |
| FAM_78 | 50 | 56 | 0 | 0 | 0 | 0 | 34 | 22 | 0 | 0 | 0 |
| FAM_1402 (TIR *Thon*) | 693 | 53 | 0 | 6 | 8 | 5 | 0 | 0 | 3 | 16 | 15 |
| FAM_1257 (TIR *Merou*) | 522 | 49 | 0 | 0 | 5 | 5 | 3 | 2 | 7 | 12 | 15 |
| FAM_670 | 46 | 44 | 0 | 0 | 9 | 1 | 2 | 5 | 2 | 24 | 1 |
| FAM_2649 | 762 | 40 | 0 | 0 | 16 | 24 | 0 | 0 | 0 | 0 | 0 |
| FAM_1473 | 98 | 33 | 0 | 0 | 4 | 7 | 1 | 0 | 5 | 10 | 6 |
| FAM_51 | 231 | 32 | 0 | 0 | 0 | 0 | 12 | 9 | 11 | 0 | 0 |
| FAM_692 | 93 | 26 | 0 | 0 | 4 | 13 | 5 | 4 | 0 | 0 | 0 |
| FAM_1294 (*Baudroie*) | 1,706 | 26 | 0 | 0 | 0 | 0 | 1 | 2 | 18 | 4 | 1 |
| FAM_2314 (DDE) | 3,421 | 24 | 0 | 20 | 2 | 2 | 0 | 0 | 0 | 0 | 0 |
| FAM_2802 | 32 | 22 | 0 | 0 | 2 | 20 | 0 | 0 | 0 | 0 | 0 |
| FAM_3194 | 230 | 20 | 20 | 0 | 0 | 0 | 0 | 0 | 0 | 0 | 0 |
| FAM_837 | 50 | 18 | 0 | 0 | 0 | 0 | 14 | 4 | 0 | 0 | 0 |
| FAM_1165 | 77 | 15 | 0 | 0 | 0 | 0 | 1 | 1 | 0 | 12 | 1 |
| FAM_2936 | 223 | 15 | 0 | 0 | 15 | 0 | 0 | 0 | 0 | 0 | 0 |
| FAM_1259 | 231 | 14 | 0 | 0 | 0 | 0 | 1 | 0 | 13 | 0 | 0 |
| FAM_3023 | 350 | 11 | 0 | 0 | 11 | 0 | 0 | 0 | 0 | 0 | 0 |
| Total | | 7,443 | 21 | 147 | 4,095 | 1,109 | 419 | 371 | 827 | 224 | 230 |

exceptions are FAM_1257, FAM_1402, and FAM_78 elements, which are underrepresented within genes (S4 Table). In particular, FAM_78 elements are exclusively found in intergenic regions.

As explained previously, it is possible to date insertions for the subset of IESs located within well-conserved protein-coding regions. The vast majority (97.5%) of mobile IES copies that can be dated correspond to recent insertions (as compared to only 9.5% of recent insertions for the other IESs). FAM_3 is present in all genomes of the subclade A (Table 2), and 94% of dated insertions are shared by at least 2 species, which indicates that this element has been very active at the beginning of the radiation of this clade. For the other families of mobile IESs, more than 97% of insertion loci are species specific. Thus, all the families of mobile IESs that we detected are relatively recent, posterior to the radiation of the *aurelia* clade. This most probably reflects the fact that more ancient families are difficult to recognize, because of the rapid divergence of IES sequences.

In agreement with previous analyses [4], we detected few cases of recent mobilization of IESs in *P. tetraurelia* (although we detected more copies than reported by [4], owing to the higher sensitivity of the sequence similarity search protocol used here). However, several other species contain IES families with a high number of recently inserted copies (Table 2). The

largest family (FAM_2183) corresponds to a 233-bp-long nonautonomous element, for which we detected a total of 4,126 copies in *P. sonneborni*, and 1,089 copies in *P. tredecaurelia*, among which 20% are nested within IESs and 80% correspond to mobile IESs (Table 2, Fig 2B–2D). The high level of sequence similarity between all copies (average pairwise identity = 72%; Fig 2) suggests that both species have been invaded recently by this mobile element. To assess whether some insertions are ancestral to these 2 species, we analyzed copies inserted in well-conserved coding regions (i.e., for which it is possible to check whether they are co-orthologous or not; *N* = 1,973 copies). We identified only 2 FAM_2183 elements inserted at co-orthologous sites in both species. Thus, the quasi-totality of FAM_2183 copies result from independent waves of insertion in the 2 lineages. Interestingly, FAM_2183 is absent from other *Paramecium* species (except 4 copies in *P. sexaurelia*). It is important to note that *P. sonneborni* and *P. tredecaurelia* belong to 2 distantly related subclades of the *aurelia* complex (Fig 1). This patchy phylogenetic distribution therefore suggests that FAM_2183 has been subject to horizontal transfers between *P. sonneborni* and *P. tredecaurelia* lineages. The alternative hypothesis of vertical inheritance of the family since the last common ancestor of the *aurelia* complex would imply that all copies of the family have been lost independently in at least 3 lineages of subclade A. Furthermore, the phylogeny of this family is mainly star-like, with *P. sonneborni* and *P. tredecaurelia* copies interspersed across the entire tree (Fig 2B). This topology is incompatible with a scenario of vertical inheritance, in which copies from each species would have been expected to form 2 distant clades. Interestingly, the tree topology indicates that several events of horizontal transfer, not just one, occurred between these 2 lineages. This finding is corroborated by 4 other families (FAM_2317, FAM_2321, FAM_2649, and FAM_2802) that are shared specifically by *P. tredecaurelia* and the *P. sonneborni*/*P. sexaurelia* clade. These multiple events of horizontal transfer suggest an important genetic flux between these 2 lineages. And indeed, besides these mobile IESs, we also found many large genomic segments (up to 200 kb) that appear to have been introgressed into MIC-specific regions of *P. sonneborni* genome, from a lineage closely related to *P. tredecaurelia* (these results will be described in detail in another article). The hypothesis that we propose is that these introgressed segments brought some active TEs (both autonomous or nonautonomous), including some mobile IESs, which then were able to proliferate within the *P. sonneborni* genome. The finding of 2 FAM_2183 copies present at co-orthologous sites is a priori unexpected under this scenario. However, they both correspond to nested copies. Thus, one possible explanation is that, by chance, these copies were inserted independently in both lineages within preexisting co-orthologous IESs. An alternative hypothesis is that both nested insertions might have been transferred between species through the introgression of larger genomic segments. Like most species of the *aurelia* complex, *P. sonneborni* and *P. tredecaurelia* have a worldwide geographic distribution [24,26]. Our observations indicate that despite the strong reproductive isolation between extant species [26], horizontal transfers did occur recently between these genetically very distant lineages.

## IES excision mechanism varies with IES age

Like any biological process, the excision of IESs during new MAC development is not 100% efficient [4,29]. For example, the IES retention rate in *P. tetraurelia* MAC chromosomes is on average 0.8% in wild-type cells [30]. We observed that a substantial fraction of IESs have a much lower excision efficiency. In all *Paramecium* species, the proportion of "weak" IESs (defined as IESs with more than 10% retention in wild-type cells) differs strongly among genomic compartments: from 0.7% on average for IESs located within genes (introns or exons) to 5.4% for IESs in intergenic regions (S10 Fig). This difference probably results from the fact

that IESs with low excision efficiency are more deleterious, and therefore more strongly counterselected, in genes than in intergenic regions. Interestingly, we also observed that within coding regions, the proportion of weak IESs is much higher for newly gained IESs (2.1% on average) than older ones (0.3%) (S10 Fig). This suggests that after a wave of insertions, genomes accumulate changes in *cis* (within IESs) and/or in *trans* (in the IES excision machinery) that progressively make these new IESs more efficiently excised, presumably in response to the selective pressure against retention of IESs within coding regions.

In *P. tetraurelia*, functional analyses have revealed that different classes of IESs rely on different excision pathways [30–32]. A large subset of IESs (70%) requires the histone H3 methyltransferase Ezl1 for excision [30]. A much smaller subset (7%), all among the *EZL1*-dependent IESs, also requires the Dcl2/3 proteins, which are necessary for the biogenesis of 25 nt long scnRNAs [30,33]. The remaining 30% of IESs require neither Ezl1 nor Dcl2/3 to complete excision. Using published IES excision efficiency datasets upon silencing of *EZL1* or *DCL2/3* [30], we found that 92% of newly inserted *P. tetraurelia* IESs are sensitive to Ezl1, as compared to 39% for old ones (Fig 3). Similarly, the proportion of Dcl2/3-dependent IESs varies from 17% for new IESs to 3% for old ones. These observations suggest that newly inserted IESs, like TEs themselves [34], initially depend on histone marks deposited by Ezl1 (and to some extent on the scnRNA pathway). Over time, histone marks and scnRNAs become dispensable as IESs gradually acquire features that allow them to be efficiently excised.

We also compared the length of IESs according to their age. As previously observed for *P. tetraurelia* [4], IESs have a characteristic length distribution with the same approximately 10 bp periodicity in all *aurelia* species (Fig 4A), likely reflecting structural constraints on the excision process [4,7]. We observed that the length distribution of IESs changes drastically over evolutionary time. For example, in *P. sonneborni*, *P. tredecaurelia*, and *P. tetraurelia*, the proportion of IESs in the first peak of the length distribution (<35 bp) ranges from 1% to 10% for new IESs to 81% to 84% for old ones (Fig 4B), and similar patterns are observed in all other *aurelia* species (S11 Fig). In *P. caudatum*, the overall length distribution is shifted toward shorter IESs (71% in the first peak, compared to 35% in *aurelia*; Fig 4A). This suggests that this lineage has not been subject to IES insertion waves for a long period of time, in agreement with the paucity of recognizable mobile IESs in that genome (Table 2).

## Genomic distribution of IESs according to their age

Because of the rapid divergence of noncoding sequences, it is generally not possible to assess homology among IES insertion sites located in intergenic regions, and, hence, it is not possible to date them directly. We therefore used the length of IESs as a rough proxy for their age to investigate their genomic distribution over time. We observed that in all *aurelia* species, long IESs (>100 bp, presumably young) are uniformly distributed across genomic compartments (introns, coding regions, and intergenic regions) (S12 Fig). Conversely, short IESs (<35 bp, presumably older) are enriched in coding regions (on average, 81% of short IESs in coding regions versus 70% expected; S12 Fig). This suggests that IESs located within intergenic regions have a shorter life span than those located in coding regions.

## Evidence that some IESs are functional

A large majority of detected IESs predate the divergence of the *aurelia* clade (Fig 1). Because of the rapid evolution of noncoding sequences, co-orthologous IESs from different species are generally too divergent to be recognized by sequence similarity search. Yet the comparison of all sequences against each other revealed several interesting exceptions. Overall, we identified 69 clusters of co-orthologous IESs conserved across at least 5 of the 8 species of the *aurelia*

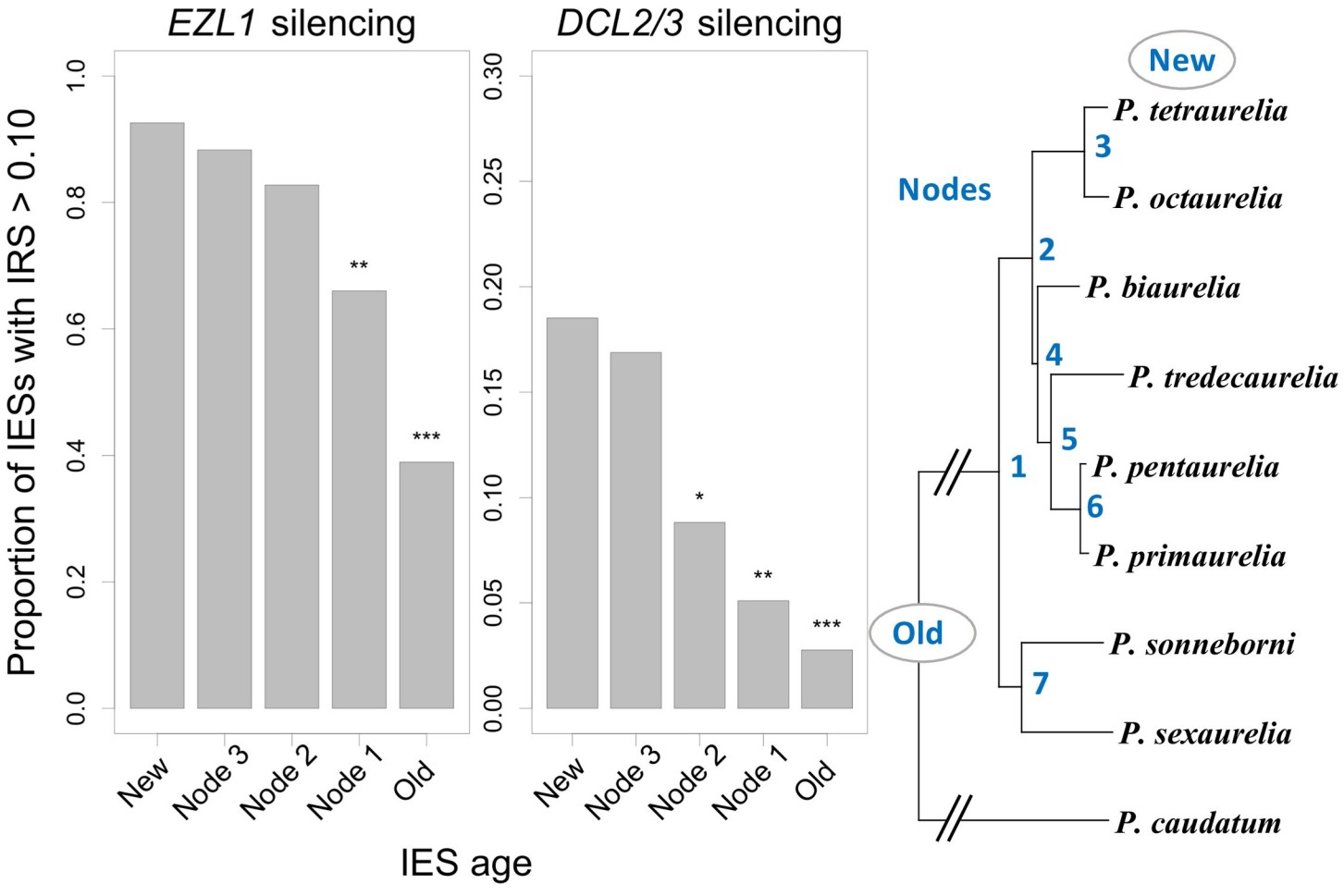

**Fig 3. Old and young IESs rely on different excision pathways.** The subset of IESs whose excision relies on histone marks or scnRNAs have been identified in *P. tetraurelia* by measuring their IRS upon silencing of *EZL1* and *DCL2/DCL3* [30]. Barplots represent the proportion of sensitive IESs (IRS > 10%) according to their age, for each silencing experiment. The age of an IES insertion is defined by the phylogenetic position of the LCA of species sharing an IES at the same site (New: *P. tetraurelia*-specific IES; Node *n*: The LCA corresponds to node number *n* in the species phylogeny; Old: The LCA predates the *P. aurelia*/*P. caudatum* divergence). The proportion of sensitive IESs among new IESs was compared to that of older ones by a chi-squared test (*: *p*-value < 0.05; **: *p*-value < 1e-3; ***: p-value < 1e-6). The data underlying this figure may be found at https://doi.org/10.5281/zenodo.4836464. IES, internal eliminated sequence; IRS, IES retention score; LCA, last common ancestor.

clade and showing at least 70% identity between sequences from subclades A and B (87% identity on average). Each of these 69 IESs correspond to unique sequences (we did not detect any interspersed homologs within genomes, only co-orthologs). These highly conserved IESs are similar to other IESs in terms of length (mean = 75 bp) or genomic distribution (79% within protein-coding genes, 21% in intergenic regions). Their high levels of sequence conservation indicate that they are subject to strong selective constraints and hence that they have a function beneficial for *Paramecium*. By definition, IESs are absent from the MAC genome so they cannot be expressed in vegetative cells. However, they can potentially be transcribed during the early development of the new MAC, before IES excision occurs [31,35]. To gain insight into their possible functions, we analyzed the transcription of conserved IESs using polyadenylated RNA-Seq data from autogamy time course experiments in *P. tetraurelia* [36]. Among the 56 families of highly conserved IESs present in *P. tetraurelia*, 10 (18%) are transcribed at substantial levels (>1 RPKM) during autogamy (as compared to 0.8% for other IESs) (S5 Table). One of these IESs (approximately 800 bp long) contains a gene encoding a putative DNA-binding

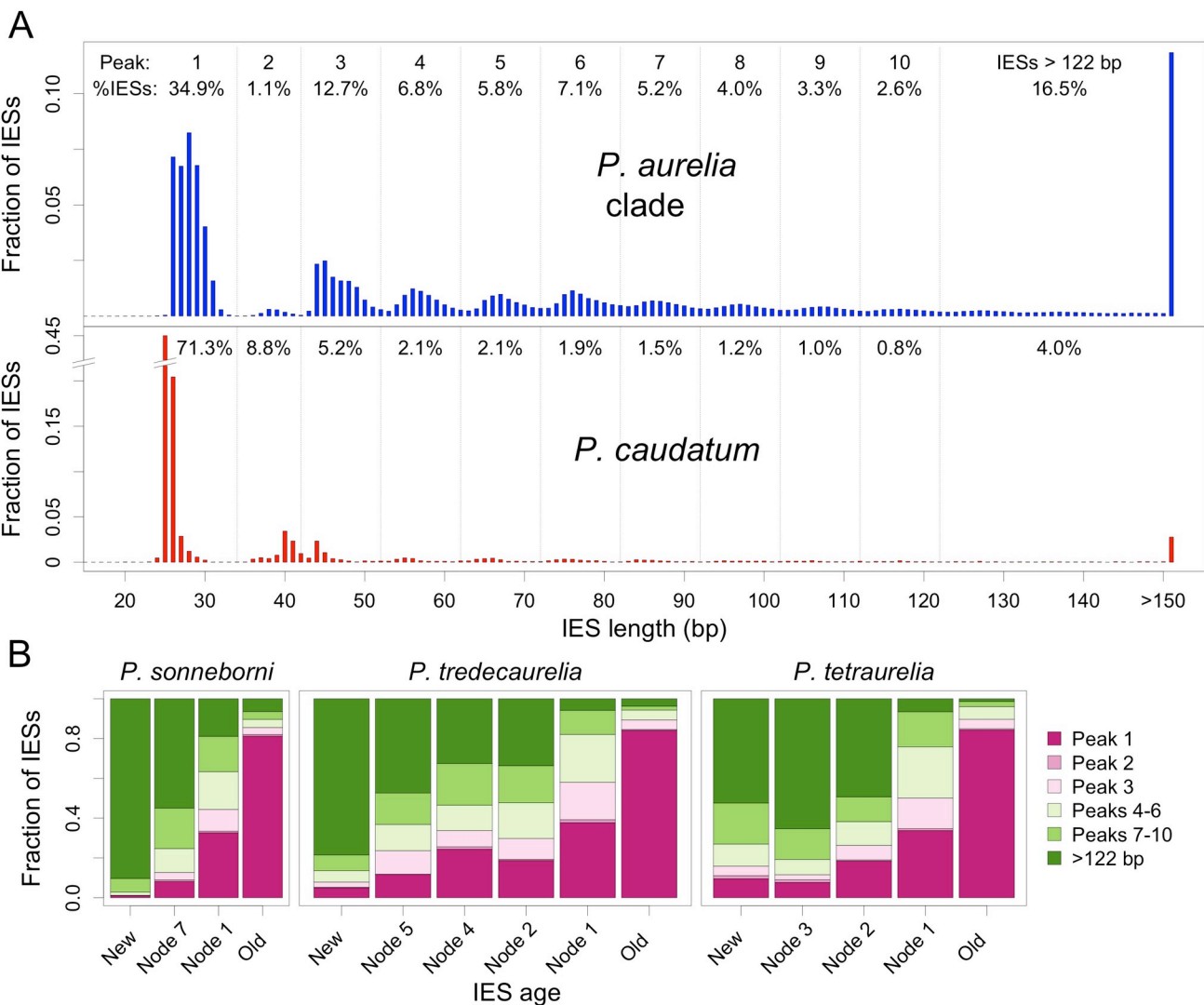

**Fig 4. Length distribution of IESs according to their age.** (A) Comparison of the length distribution of IESs in *P. caudatum* (*N* = 8,172 IESs) and in species from the *aurelia* clade (*N* = 392,082 IESs). The fraction of IESs present within each peak of the distribution is indicated for the first 10 peaks. (B) Comparison of the length distribution of IESs according to their age (for the subset of datable IESs located in coding regions). The age of IES insertions is defined as in Fig 3. Results from other species are presented in S11 Fig. The data underlying this figure may be found at https://doi.org/10.5281/zenodo. 4836464. IES, internal eliminated sequence.

protein, well conserved in all species of the *aurelia* clade and expressed at high levels during the early stages of autogamy (Fig 5A and 5C). Several paralogs of this gene are present in the MAC genome, but it is the only member of this family to be located within an IES. We hypothesize that this copy might have originated by retrotransposition of one of the MAC paralogs and that its insertion within a preexisting IES might have been exapted, as a way to restrict its expression specifically to early MAC development.

All other highly conserved IESs are much shorter (<300 bp), most probably too short to encode proteins. But we found examples suggesting that some of them contribute to the regulation of the expression of their host gene. For example, we identified a conserved IES located at the 5′ end of a gene of unknown function, encompassing the transcription start site and the

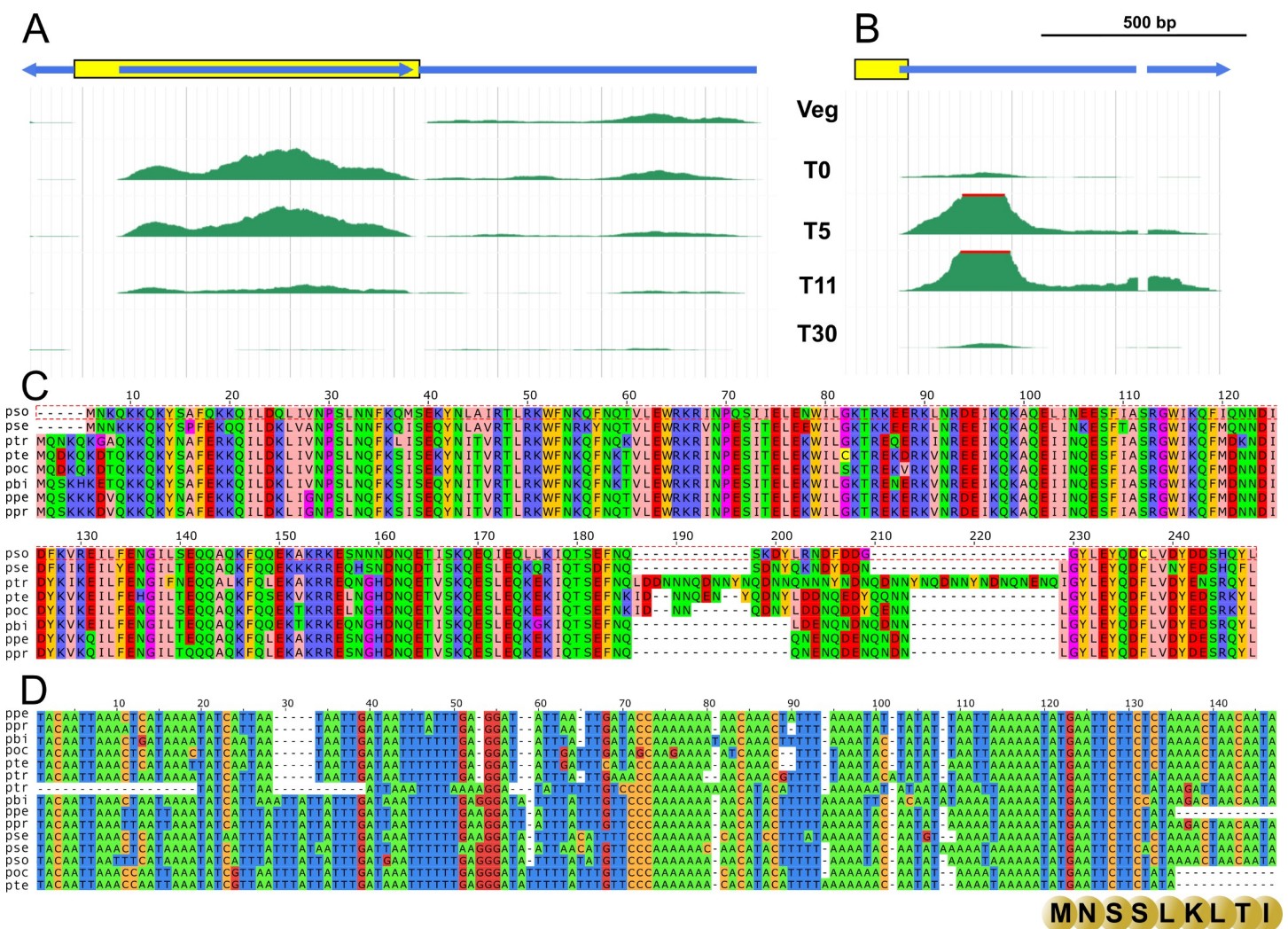

**Fig 5. Examples of highly conserved co-orthologous IESs.** (A) Cluster 4968 corresponds to a set of co-orthologous IESs that contain a gene (blue), oriented in the opposite direction with respect to the gene (blue) in which the IES is inserted (yellow). The IES-embedded gene encodes a protein belonging to a family of CENP-B DNA-binding proteins and is likely of cellular origin. RNA-Seq data (shown here for the *Paramecium tetraurelia* member of this cluster: IES.PTET.51.1.16.324097) show that this gene is expressed at meiosis, with mRNA steady-state levels strongly decreasing before most genome-wide IES excision occurs [36]. (B) Cluster 9405 corresponds to a set of co-orthologous IESs that contain the promoter and the start codon of a gene (blue) encoding a protein of unknown function. The IES is drawn in yellow. RNA-Seq data (shown here for 1 *P. tetraurelia* member of this cluster: IES.PTET.51.1.28.104046) indicate that this gene is expressed at the time genome rearrangements occur, with a peak in steady-state mRNA level at T11, when IES excision peaks [36]. (C) Alignment of the proteins encoded by IESs from cluster 4968. (D) Nucleotide alignment of co-orthologous IESs from cluster 9405. The ATG start codon and first few amino acids encoded by the IES are shown below the alignment. [IES IDs in the alignments, from top to bottom (C): IES.PPENT.87.1.080.374281, IES.PPRIM.AZ9-3.1.072.352290, IES.PBIA.V1_4.1.0053.93462.rc, IES.POCT.138.1.076.348969, IES.PTET.51.1.61.348482, IES.PTRED.209.1.7180000129326.89420.rc, IES.PTRED.209.1.7180000129337.349035, IES.PBIA.V1_4.1.0022.293968, IES.PPENT.87.1.047.444829, IES.PPRIM.AZ9-3.1.042.468625, IES.PSEX.AZ8_4.1.047.365473, IES.PSEX.AZ8_4.1.067.78433.rc, IES.PSON.ATCC_30995.1.070.107189.rc, IES.POCT.138.1.047.443628, IES.PTET.51.1.28.104046.rc. (D): IES.PSON.ATCC_30995.1.014.460171, IES.PSEX.AZ8_4.1.004.493121, IES.PTRED.209.1.7180000129358.256806, IES.PTET.51.1.16.324097, IES.POCT.138.1.015.320413, IES.PBIA.V1_4.1.0089.86544, IES.PPENT.87.1.022.397764, IES.PPRIM.AZ9-3.1.019.378988.]. IES, internal eliminated sequence.

beginning of the first exon (including the 5′UTR and the first 9 codons). The excision of the IES during MAC development leads to the loss of the initiation codon and of the promoter region, and thereby to the silencing of this gene in vegetative cells (Fig 5B and 5D). These examples illustrate that recruitment of the IES excision machinery during evolution to contribute new functions beneficial for *Paramecium*.

## Discussion

### A majority of *Paramecium* IES insertions result from the transposition of mobile IESs

To explore the evolutionary origin of IESs, we analyzed the MIC genomes of 8 species of the *P. aurelia* complex, and of an outgroup species, *P. caudatum*. Unexpectedly, we discovered that the MIC genomes of *P. caudatum* strains are at least 1 order of magnitude larger than those of *P. aurelia* species (approximately 1,600 to 5,500 Mb versus approximately 110 to 160 Mb). The sequencing of *P. caudatum* My43c3d revealed that its huge MIC genome size is caused by the amplification of 2 major satellite repeats, which represent 71% of its MIC-limited genome (S4 Fig). The high variability of genome sizes across the *P. caudatum* lineage makes this clade an attractive model system to study the possible phenotypic consequences of genome size variations within a species.

All the *Paramecium* MIC genomes we sequenced present a high density of IESs in MAC-destined sequences: from 0.5 IES per kb in *P. caudatum* up to 1 IES per kb in *P. sonneborni* (Table 1). The vast majority of these IESs (83% on average) are located within genes, as expected given the very high gene density in MAC genomes (S3 Table). In *aurelia* species, there are on average 0.95 IESs per protein-coding gene. The IES density varies among genes, but overall, approximately 50% of the approximately 40,000 genes contain at least 1 IES. Moreover, the analysis of MIC-specific regions that are occasionally retained in the MAC (MAC-variable regions) revealed similar IES densities (S2 Table), which suggests that, in addition to IESs located in MAC-destined regions, many other IESs are located within MIC-specific regions.

To explore the origin and evolution of these tens of thousands of IESs, we sought to identify homologous IESs across the 9 *Paramecium* species. We distinguished 2 categories of homologs: interspersed homologs (i.e., similar IES sequences located at nonhomologous loci) and co-orthologous IESs (i.e., IESs deriving from a single ancestral insertion event at a given genomic site). We first compared all IES against each other to identify sets of homologs based on sequence similarity, within genomes or across species. The vast majority of IESs correspond to sequences that are unique within each genome (Fig 2). We identified only a handful of co-orthologous IES sequences that are conserved in sequence across the *aurelia* complex (see below). Apart from these rare cases, most IES sequences evolve rapidly, as expected for non-coding sequences. It should be noted that species from the *aurelia* complex are genetically quite divergent (S6 Table). For example, the average synonymous divergence (measured in orthologous protein-coding genes) is 0.95 substitutions/site between species of the A and B subclades. At this evolutionary scale, in the absence of selective pressure, IES homologs are expected to be far too diverged to be recognizable by sequence similarity. Thus, homologs (interspersed or co-orthologs) that can be detected by sequence similarity essentially correspond to IESs whose last common ancestor is relatively recent (more recent than the A and B subclades).

However, it is also possible to identify co-orthologous IESs based on their shared position within multiple alignments of homologous genes. Thus, for the subset of IESs located in coding regions, we were able to infer rates of IES gain and loss across the species phylogeny (Fig 1). Overall, about 90% of IESs detected in *aurelia* species predate the radiation of that clade, but fewer than 10% are shared with *P. caudatum*. Thus, the vast majority of *aurelia* IESs result from a major wave of IES gains that occurred after the divergence of *P. caudatum*, but before the radiation of the *aurelia* complex. Similarly, 80% of IESs detected in *P. caudatum* are specific to that lineage, which implies that multiple independent events of massive IES invasions occurred during evolution.

The burst of IES gains at the base of the *aurelia* clade was followed by a progressive slowdown in most species, except in the *P. sonneborni* lineage, which has been subject to a second wave of IES insertions (Fig 1). Interestingly, the comparison of IES sequences revealed thousands of interspersed homologous copies, resulting from the recent and massive mobilization of a small number of IESs. Several families of mobile IESs present homology with known ITm transposons, and some of them encode transposases. But most mobile IESs do not appear to have any protein-coding potential and therefore likely correspond to nonautonomous elements, whose mobility depends on the expression of active transposons. The number of detectable (recently mobilized) IES copies varies widely across species (Table 2). For example, mobile IESs have been very active in the *P. sonneborni* lineage (4,095 copies), much more than in its sister lineage, *P. sexaurelia* (147 copies). Thus, mobile IESs account for at least 20% of the difference in IES number between these two species (Fig 1). Given the rapid evolution of IESs, interspersed homologs that can be detected probably represent only the tip of the iceberg of mobile IESs. Overall, we found a strong correlation ($R^2 = 0.86$, $p = 8 \times 10^{-4}$) between the number of mobile IES copies detected in each species (Table 2) and the rate of IES gain along corresponding branches of the phylogeny (Fig 1), which suggests that most gains result from transposition.

Interestingly, the 5 most active families in *P. tredecaurelia* all show the signature of horizontal transfer with the distantly related *P. sonneborni* lineage (Table 2). This is the case of the largest family that we identified (FAM_2183: 3,252 and 897 copies in each species, respectively; Fig 2). This pattern is reminiscent of the typical life cycle of many DNA transposons: When a new element enters a genome, it is initially very active and produces a wave of insertions. Its activity then progressively slows down, largely because defense mechanisms become more efficient in the host genome. In the long-term, DNA transposons escape extinction only if they can occasionally be transmitted to a new host [37]. Thus, the variation of IES insertion rates that we observed in the *Paramecium* phylogeny fits very well with the dynamics of TEs: rare episodes of massive proliferation (promoted by horizontal transfer of an active copy to a new naive host), followed by progressive slowdown of transposition activity.

TEs are not the unique source of IES gains. Mutations in MAC-destined regions can generate sequence motifs that are recognized by the IES excision machinery and thereby create new IESs. There is indeed evidence that cryptic IES signals occasionally trigger the excision of MAC-destined sequences [4,29] and that some IESs originated from MAC-destined sequences [12,13]. However, our results suggest that the vast majority of IESs correspond to unrecognizable fossils of mobile elements—as initially proposed by Klobutcher and Herrick [10,11].

## Indirect evidence of mutational burden caused by IES invasions

In all *Paramecium* species, we observed a deficit of IESs in highly expressed genes (S7 Fig). As previously reported in *P. tetraurelia* [4], this pattern most probably reflects selective pressure against IES insertions within genes. Indeed, the IES excision machinery (like any other biological machinery) is not 100% efficient: A small fraction of IES copies are retained in the MAC or subject to imprecise excision [29]. Typically, the average IES retention rate in MAC chromosomes is 0.8% in *P. tetraurelia* [30]. For IESs located within genes, such excision errors are expected to have deleterious consequences on fitness, in particular for genes that have to be expressed at high levels [4]. And indeed, in agreement with the hypothesized selective pressure against IESs within genes, we observed that the proportion of "weak" IESs (i.e., IESs with a relatively high retention frequency) is much lower in genes than in intergenic regions (S10 Fig).

Despite their selective cost, weakly deleterious IES insertions can eventually become fixed by random genetic drift. Once fixed, the fitness of the organism will depend on its ability to

properly excise the IES during MAC development. Over time, selection should favor the accumulation of substitutions and indels that increase the efficiency of IES excision. Indeed, we did observe that the proportion of weak IESs decreases with their age (S10 Fig). Interestingly, older IESs, which are also shorter, are less dependent on the Ezl1 and Dcl2/3 proteins (Fig 3). This suggests that after their insertion, IESs progressively acquire features that make them more efficiently excised, by a pathway that requires neither scnRNAs nor histone marks [30].

## Exaptation of the IES excision machinery

As mentioned previously, orthologous sequences that evolve neutrally are not expected to display any significant similarity between species of subclades A and B (synonymous divergence approximately 0.95 substitution per site; S6 Table). Yet, we identified 69 clusters of co-orthologous IESs displaying a high level of sequence conservation across the entire *aurelia* complex (on average 87% identity between the closest pair of homologs from subclades A and B). Their level of conservation implies that these IESs are subject to strong purifying selection and hence that they fulfill a function that contributes to the fitness of *Paramecium*. Notably, we identified 1 IES that contains a protein-coding gene (Fig 5A and 5C). This gene is expressed during the early stages of autogamy, likely from the new developing MAC, before IES excision (Fig 5A). Interestingly, 18% of the conserved IESs are transcribed during autogamy (as compared to 0.8% for other IESs). Most conserved IESs are too short to encode proteins, but they may contribute to gene regulation (e.g., Fig 5B and 5D). Given the enrichment of conserved IESs in genes expressed during early autogamy, it is tempting to speculate that these IESs may play a role in controlling the IES excision machinery itself. Indeed, this machinery must be tightly regulated to ensure that all IESs are efficiently excised, while limiting off-target excision of MAC-destined regions, which occurs occasionally in MAC chromosomes [4,29]. Thus, developmental disruption of genes encoding IES excision factors by the excision machinery may provide a simple regulatory feedback loop to decrease the activity of the IES excision machinery as soon as a large fraction of IESs have been excised: If a given IES drives the expression of a protein factor that is essential for IES excision, then this process is progressively interrupted by the removal of this IES during MAC development. More generally, such IESs may provide an exquisite developmental process to regulate DNA elimination events and/or MAC differentiation.

Given that each of these 69 IESs is conserved across species and unique within genomes, it seems a priori unlikely that they derive from TEs. We propose that these conserved IESs may in fact originate from MAC-destined segments. Indeed, mutations within genes can create sequence motifs recognized by the IES excision machinery and thereby trigger the deletion of functional elements during MAC development. Such mutations are expected to be generally deleterious and hence to be counterselected. But occasionally, they might have been positively selected in some genes because they provide a mean to modulate their expression during MAC development or to generate phenotypic diversity among clones, as previously observed for IESs overlapping mating-type genes in many species [12,13]. This scenario would notably explain the presence of conserved IESs overlapping with cellular genes (e.g., Fig 5B). Thus, we propose that the IES excision machinery, which evolved initially to ensure the efficient removal of selfish genetic elements from the MAC genome, has been exapted during evolution as a new way to regulate the expression of some genes.

## IES losses

In all species, we observed that the length of IESs is negatively correlated with their age (Figs 4 and S11). This pattern is similar to that observed in other eukaryotes, where fixed copies of

TEs tend to shrink over time, due to the accumulation of small deletions [38]. For IESs located in noncoding region, this progressive shrinking can ultimately lead to their disappearance. Furthermore, these IESs can also be lost by transformation into a MAC-destined sequence (e.g., via mutations the TA dinucleotides). However, for IESs located within exons, losses can only occur by precise complete deletions that leave the ORF intact. We did observe such cases of precise loss (S9 Fig). One possible mechanism is that the IES excision machinery, which is normally at work during MAC development, might occasionally operate within the MIC or zygotic nucleus. An alternative hypothesis is that IESs might be lost from the MIC lineage by gene conversion through homologous recombination with MAC-derived DNA fragments. Interestingly, this scenario might explain cases where we observed concomitant losses of neighboring IESs (see, e.g., IES 5 and 6 in S9 Fig). Further studies will be needed to determine the mechanisms underlying precise IES loss. The rate of IES loss has remained quite stable and relatively low across the phylogeny (Fig 1). Conversely, the rate of IES gains has been much more erratic, characterized by episodic waves of insertions, during which the IES gain rate largely exceeded the loss rate (Fig 1). In the end, the number of IESs reflects the balance between gain and loss rates. Thus, the large number of IESs in *Paramecium* can simply be explained by massive invasions of mobile IESs, followed by periods of lower activity, during which IES copies progressively diverge, and occasionally get lost by deletion from the MIC.

## Scenario for the evolution of IESs

In most organisms, gene regulatory elements and coding regions constitute a no man's land for TEs, because insertions that disrupt gene function are strongly counterselected. But in some ciliates, it is possible for mobile elements to proliferate within genes in the MIC genome, as long as they are efficiently and precisely excised during the development of the MAC genome, before genes start to be expressed. DNA transposons encode transposases that allow their mobilization by a "cut-and-paste" process. Generally, the excision step leaves a few nucleotides at the original insertion site, but one peculiarity of piggyBac transposases is that they can excise copies precisely, without leaving any scar [39]. This feature may have predisposed piggyBac to extend its niche to genic regions in ciliates. We speculate that the very first proto-IESs corresponded to piggyBac elements whose transposase was domesticated to target the developing MAC. As soon as several copies of these proto-IESs have been fixed within genes, then the host organism has become dependent on the activity of the piggyBac transposase to ensure that all these copies are precisely excised from its MAC. This selective pressure would have perfected the domestication of the piggyBac transposase by its host, and then, progressively, the evolution of the other components that contribute to the efficient excision of proto-IESs. Once the IES excision machinery in place in the ancestral *Paramecium* lineage, other families of TEs (including nonautonomous elements) could hijack the machinery and in turn exploit this intragenic niche, eventually creating the tens of thousands of IESs found in present-day *Paramecium* genes. The first steps of this scenario remain speculative, since there are no recognizable traces of piggyBac-related IESs in present-day genomes. But the discovery of thousands of mobile IESs directly demonstrates the major contribution of TEs to the expansion of the IES repertoire.

The coexistence of MAC and MIC is a common feature of all ciliates, yet they do not all contain such a high density of IESs in coding regions. Notably, there are approximately 12,000 IESs in the germline genome of *Tetrahymena thermophila* (approximately 0.1 IES per kb of MAC-destined sequence), but only 11 of them are located within coding regions [40]. These exonic IESs differ from other IESs by their strongly conserved TIRs ending with 5′-TTAA-3′, the target site of piggyBac transposons. They are excised precisely (restoring a single TTAA)

by 2 domesticated piggyBac transposases, Tpb1 and Tpb6, which may thus have retained the cleavage specificity of their transposon ancestor [41,42]. We analyzed these 11 exonic IESs: 8 of them are inserted in protein-coding regions that are not conserved in *Paramecium*, and the other 3 are inserted at sites that do not contain IESs in *Paramecium*. There is therefore no evidence for shared exonic IESs between *T. thermophila* and *Paramecium*. The vast majority of the approximately 12,000 *T. thermophila* IESs are excised by another domesticated piggyBac transposase, Tpb2 [43]. Although Tpb2 retains the cleavage geometry of piggyBac transposases, producing staggered double-strand breaks with 4-nt 5′ overhangs [43], it has lost almost all sequence specificity and is thought to be recruited at IES ends by chromatin marks [44]. As a result, several possible cleavage sites are usually present at IES ends and the rejoining of flanking sequences generates microheterogeneity in the MAC sequence [40], which explains why Tpb2-dependent IESs are restricted to introns and intergenic regions [40]. It is important to note that Tpb2 is an essential gene in *T. thermophila* [43], suggesting that genome-wide retention of IESs in the MAC is still highly detrimental. Interestingly, phylogenetic analyses indicate that the *Paramecium* endonuclease Pgm and Tpb2 are more closely related to each other than to Tpb1 or Tpb6 and may even be orthologs [7]. In the case of Pgm, however, sequence specificity was relaxed only for the 2 distal positions of the 4-nt cleavage sites, and the central TAs remain a strict requirement for IES excision in *Paramecium*. Although piggyBac transposons are completely absent from the present-day *Paramecium* germline, this evolutionary solution may have been favored because it also allowed for precise excision of Tc1/mariner insertions, which, in turn, would have allowed continuous accumulation of insertions within exons [4].

Importantly, the fact that a mechanism of precise excision exists in *T. thermophila* (via Tpb1 and Tpb6) raises the question of why intragenic IESs are not more abundant in its genome. A similar question arises from the distribution of introns in eukaryotes: Why are introns very abundant in some lineages but rare in others (e.g., approximately 7 introns per gene in vertebrates versus approximately 0.04 in hemiascomycetous yeast)? Part of the explanation may reside in the fact that, because of population genetic forces, some lineages are more subject to random genetic drift than others and therefore are more permissive to invasion by weakly deleterious genetic elements [45,46]. It is also possible that the abundance of intragenomic parasites is strongly affected by contingency—rare events of massive invasion, followed by long periods during which copies are lost at a slow rate.

## Parallel scenario for the evolution of IESs and spliceosomal introns

The above scenario for the origin of IESs in *Paramecium* presents an interesting parallel with the one proposed for the evolution of spliceosomal introns in eukaryotes. Indeed, it had long been postulated, based on similarities in biochemical processes, that spliceosomal introns derive from mobile elements (group II self-splicing introns) [47]. In eukaryotes, the spread of introns in protein-coding genes has been facilitated by the fact that transcription and translation occur in separate compartments, thus offering the opportunity for these mobile elements to be excised from the mRNA in the nucleus without interfering with its translation in the cytoplasm [47]—like IESs, which are excised from genes before they get expressed in the MAC. Once the first introns were established, selection drove the emergence of host factors contributing to the efficiency of the splicing process, which progressively led to the evolution of the modern spliceosome [48]. In turn, the existence of the spliceosome released the requirement for introns to maintain their self-splicing activity [49] and allowed other TEs to hijack this machinery. The recent discovery of nonautonomous DNA transposons that generated thousands of introns in genomes of algae directly demonstrated that mobile elements are a

major source of new introns [50]. Like IESs, introns represent a burden for their host, because of errors of the splicing machinery [45,46, 51–53], but they also contributed to innovations. Indeed, the spliceosome has been exapted during evolution to produce alternative splicing, which not only contributed to diversify the protein repertoire [54] but also allowed new modes of posttranscriptional regulation gene expression [55], in particular in genes that encode splicing factors [56,57]. This pattern is reminiscent of highly conserved IESs that we uncovered in *Paramecium* lineages, which appear to be particularly enriched within genes that are expressed during early MAC development.

In summary, the evolution of the nuclear envelope opened the way for introns to invade genes in eukaryotes [47], and likewise, the separation of somatic and germline functions between the MIC and the MAC offered the possibility for selfish genetic elements to invade genes in ciliates. Genetic conflicts between these selfish elements and their host genome resulted in the evolution of complex cellular machineries (the spliceosome, the IES excision machinery), which, in the short term, reduced excision errors, but, in the long term, facilitated proliferation of these elements within genes. The paradigm of intragenomic parasites [58–60] provides a simple and powerful explanation for the "raison d'être" of these mysterious pieces of noncoding DNA that interrupt genes.

## Materials and methods

### Cells and cultivation

All experiments were carried out with the *Paramecium* strains listed in Table 1. *P. aurelia* cells were grown in a wheat grass powder (WGP, Pines International, USA) infusion medium bacterized the day before use with *Klebsiella pneumoniae* and supplemented with 0.8 mg/L of β-sitosterol. Cultivation and autogamy were carried out at 27˚C. Monoclonal cultures of the *P. caudatum* cells were grown in a 0.25% Cerophyl infusion inoculated with *Enterobacter aerogenes* at 22˚C [61].

### Micronucleus-enriched preparation

To purify the MICs from vegetative cells, we used the same strategy as the one previously published [5,19], with some optimization for the sorting steps. For *P. aurelia*, transgenic cells expressing an MIC-localized version of the green fluorescent protein (GFP) were obtained by microinjection of the vegetative MAC with the *P. tetraurelia* CenH3a-GFP plasmid, described in [62]. In the transformed clones, GFP was exclusively found in the MICs, and the transformed clones were selected for their GFP signal/noise ratio. Viability of the sexual progeny after autogamy of the transformed clones was systematically monitored to make sure that the presence of the transgene did not impair the functionality of the MICs. An MIC-enriched preparation was obtained from approximately 3 L of exponentially growing vegetative cells after fractionation and Percoll density gradient centrifugation as described in [5] and kept at −80˚C until further use.

A slightly different procedure was used for *P. caudatum* cells, which were not transformed with the CenH3a-GFP transgene. The MICs of *P. caudatum* strain My43c3d (used for genome sequencing) were purified with a protocol modified from [63]. Briefly, 3 L of a starved culture (approximately 600 cells/mL) were filtered through 8 layers of gauze and concentrated by centrifugation in pear-shaped centrifuge tubes. Packed cells were transferred to a 250-mL cell culture flask, resuspended in 150 mL sterile Eau de Volvic, and incubated over night at 22˚C. All subsequent steps were performed at 4˚C or on ice. The overnight culture was again concentrated by centrifugation, and the cell pellet was resuspended and washed in 0.25 M TSCM buffer (10 mM Tris-HCl (pH 6.8), 0.25 M sucrose, 3 mM CaCl2, 8 mM MgCl2) [64]. After

centrifugation for 3 min at 100*g*, pelleted cells were resuspended and incubated for 5 min in 10 mL 0.25 M sucrose-lysis buffer (10 mM Tris-HCl (pH 6.8), 0.25 M sucrose, 3 mM CaCl2, 1 mM MgCl2, 0.1% Nonidet-P40, 0.1% Na-deoxycholate). The cell suspension was centrifuged for 2 min at 500*g*, and the packed cells were lysed in 1 mL of 0.25 M sucrose-lysis buffer by about 10 to 20 strokes on a vortex machine. Lysed cells were washed in 14 mL of 0.25 M TSCM buffer and centrifuged for 1 min at 100*g*. The supernatant (containing the MICs) was centrifuged for 10 min at 1,500*g*, and the pellet was resuspended in 8 mL of 60% Percoll. This suspension was centrifuged for 15 min at 24,000*g* in a fixed-angle rotor, and the micronuclei formed a diffuse band near the middle of the centrifuge tube. This MIC-containing layer was carefully removed with a pipette in about 2 mL, diluted with 10 mL of 0.25 M TSCM buffer, and pelleted by centrifugation for 10 min at 1,500*g*. The MIC pellet was resuspended in 100 μL of 0.25 M TSCM buffer, carefully mixed with 50 μL of 50% glycerol, and kept at −80˚C until further use.

The MICs of the other *P. caudatum* strains were purified with a similar protocol, but omitting the Percoll step and replacing it with centrifugation across a sucrose cushion. Lysed cells were resuspended in 9 mL of 0.25 M TSCM buffer, and this suspension was carefully layered on top of a sucrose cushion consisting of 2 mL of 1.6 M TSCM buffer and 2 mL of 0.9 M TSCM buffer and centrifuged in a swinging bucket rotor for 10 min at 300*g* with lowest acceleration and braking levels. Depending on the strain, the micronuclei accumulated at the bottom of the 0.25 M or 0.9 M TSCM cushion and were removed by careful pipetting of the respective phases to new 15-mL tubes. MIC-containing suspensions were diluted with 0.25 M TSCM buffer, centrifuged for 10 min at 1,500*g*, and the MIC pellets were subsequently treated as described above.

## Quantification of MIC DNA content by flow cytometry

We measured the absolute DNA content in the nuclei with propidium iodide, a fluorophore that is insensitive to differences in base composition, and compared DNA content of MIC-enriched preparations to a standard (tomato nuclei) of known genome size (see S1 Data). MIC-enriched samples were thawed on ice, diluted 1/5 to 1/10 in washing buffer (0.25 M sucrose; 10 mM Tris (pH 7.4); 5 mM MgCl2; 15 mM NaCl; 60 mM KCl; 0.5 mM EGTA), and stained on ice with propidium iodide at 100 μg/mL final concentration. We used Tomato nuclei obtained from Montfavet 63–5 hybrid F1 seeds as internal standards of known genome size. Tomato nuclei were obtained from 1 cm$^2$ of young leaves chopped in a Petri dish with a scalpel. A volume of 800 μL of a modified Galbraith buffer [65], containing 45 mM MgCl2, 30 mM Sodium-Citrate and 20 mM MOPS (pH 7.0), 40 μg/mL RNAse A, 0.1% Triton X-100, 5 mM sodium metabisulfite ($S_2O_5Na_2$), was added. The nuclei were collected by pipetting, filtered on 70 μm mesh, and stained on ice with propidium iodide at 100 μg/mL final concentration.

The samples were analyzed on a CyanADP Cytomation analyzer from Beckman-Coulter equipped with 3 lasers: 405 nm, 488 nm, and 635 nm. Fluorescence intensity (PE signal in pulse-height) of the nuclei was measured at 575/25 nm, after excitation with the 488-nm laser. Results are deduced from 2C nuclei in individuals considered diploid and are given as C-values [66]. The ratio of fluorescence intensity of 2C-nuclei from sample and standard allows calculation of genome size. C corresponds to the nuclear genome size (the whole chromosome complement with chromosome number n), 1C and 2C being, respectively, the DNA contents of the haploid (n) and diploid (2n) sets of chromosomes. The haploid nuclear DNA content is expressed in picograms or million base pairs, where 1 pg = 978 Mbp [67], considering Tomato 2C DNA (pg) = 1.99, according to [68]. The raw data and calculations are provided in S1 Data.

## Micronucleus sorting by flow cytometry and flow imaging

To sort the MICs, the MIC-enriched samples were submitted to flow cytometry. Sorting by flow cytometry allowed the separation of the small, diploid MICs from the highly polyploid MAC and the bacteria abundant in *Paramecium* cultures. *P. aurelia* MICs were sorted based on the SSC, FSC, DAPI (DNA staining), and GFP signals. *P. caudatum* MICs, which are bigger than *aurelia* MICs, could be sorted based on their SSC, FSC, and DAPI signals, without the use of an MIC-specific GFP fluorophore. Quality control was performed by flow cell imaging, using the ImageStreamX (Amnis/Merck Millipore, France) imaging flow cytometer, as previously described [5]. The MICs represented >99% of the sorted sample, except for *P. sonneborni* (97%). An example of sorting is shown in S1 Fig.

## Genomic DNA extraction and sequencing

For MAC DNA sequencing, genomic DNA was extracted from vegetative *Paramecium* cell culture after centrifugation and washes with Tris 10 mM (pH 7.4). For MIC DNA sequencing, DNA was extracted from the sorted MIC samples. The cell or nuclei pellet was treated with 3 volumes of proteinase K solution (0.5 M EDTA (pH 9); 1% N-lauroylsarcosine; 1% SDS; 1 mg/mL proteinase K) at 55˚C overnight. Genomic DNA was extracted with Tris–HCl-phenol (pH 8) with gentle agitation followed by dialysis against TE (10 mM Tris–HCl; 1 mM EDTA (pH 8)) 25% ethanol then against Tris 1 mM (pH 8). An RNAse A treatment was performed on MAC DNA, followed by phenol extraction and dialysis as described above. DNA concentration was quantified using QuBit High sensibility kit (Thermo Fisher Scientific, France) and stored at 4˚C.

As the amounts of DNA extracted from the MIC are too low (30 to 50 ng), only an overlapping paired-end library could be prepared for de novo sequencing. Briefly, 30 to 50 ng of MIC DNA were sonicated using the E210 Covaris instrument (Covaris, USA) in order to generate fragments mostly around 500 bp. Illumina libraries were then prepared using the NEBNext DNA Sample Prep Master Mix Set (New England Biolabs, Massachusetts, USA), and DNA fragments were PCR amplified using Platinum Pfx DNA polymerase (Invitrogen) and P5 and P7 primers. Amplified library fragments of roughly 500 to 600 bp were size selected on 2% agarose gel. Libraries traces were validated on an Agilent 2100 Bioanalyzer (Agilent Technologies, USA) and quantified by qPCR using the KAPA Library Quantification Kit (KapaBiosystems) on a MxPro instrument (Agilent Technologies, USA). The libraries were sequenced using 251 base-length read chemistry in a paired-end flow cell on the Illumina HiSeq2500 sequencer (Illumina, USA).

For the MAC genomes, an overlapping paired-end library as described above, and 4 additional mate-pair libraries (about 5 kb, 8 kb, 11 kb, and 13 kb) were prepared following Nextera protocol (Nextera Mate Pair sample preparation kit, Illumina). Each library was sequenced using 100 base-length read chemistry on a paired-end flow cell on the Illumina HiSeq2000 (Illumina, USA).

Information about the sequencing data generated for this study is available in S7 Table.

## RNA extraction and sequencing

For the purpose of gene annotation, we sequenced mRNAs from vegetative cells (S7 Table). A volume of 400 mL cultures of exponentially growing cells at 1,000 cells/mL were centrifuged and flash frozen in liquid nitrogen prior to TRIzol (Invitrogen) treatment, modified by the addition of glass beads for the initial lysis step.

RNA-Seq library preparation was carried out from 1 μg total RNA using the TruSeq Stranded mRNA kit (Illumina, San Diego, California, USA), which allows mRNA strand orientation (sequence reads occur in the same orientation as antisense RNA). Briefly, poly(A)

+ RNA was selected with oligo(dT) beads, chemically fragmented, and converted into single-stranded cDNA using random hexamer priming. Then, the second strand was generated to create double-stranded cDNA. cDNAs were then 3′-adenylated, and Illumina adapters were added. Ligation products were PCR amplified. Ready-to-sequence Illumina libraries were then quantified by qPCR using the KAPA Library Quantification Kit for Illumina Libraries (Kapa-Biosystems, Wilmington, Massachusetts, USA), and library profiles evaluated with an Agilent 2100 Bioanalyzer (Agilent Technologies, Santa Clara, California, USA). Each library was sequenced using 101 bp paired-end read chemistry on a HiSeq2000 Illumina sequencer.

## MAC genome assembly

The MAC genomes sequenced for this project were all assembled according to the following steps. First, long Illumina reads were obtained from 250 bp overlapping paired-end reads sequenced from approximately 450 bp fragments. The reads were fused with fastx_mergepairs, an in-house tool developed at Genoscope using the fastx library (http://hannonlab.cshl.edu/fastx_toolkit/). An alignment of at least 15 bp with at least 90% identity and fewer than 4 errors was required to fuse 2 reads into 1 longer read. The set of fused reads, completed with any reads that could not be fused, was assembled into contigs by the Newbler version 2.9 overlap-layout-consensus assembler, with a minimal alignment identity of 99% and a minimal alignment size of 99 bp. Scaffolds were built from the contigs using 4 Illumina mate-pair libraries with respective insert sizes of 5 kb, 8 kb, 11 kb, and 13 kb. The scaffolder SSpace [69] was used, with default parameters and an acceptable variation in mate-pair insert size of 25%. Gap closing was a 2-step process with SOAPdenovo2 GapCloser software [70]. The first step used the Illumina paired-end reads; the second step used the Illumina mate-pair libraries. Finally, Kraken software [71] and the NR nucleotide database were used to detect and remove noneukaryotic scaffolds, owing mainly to bacterial contaminants (see below).

## Filtering

Scaffolds with a length inferior to 2 kb or with a G+C content greater than 40% were filtered. Contaminant scaffolds were identified and removed from the assembly provided the Kraken Kmer score was superior to 10 or a BLASTN match (-evalue 1e-40 –perc_identity 70) against RefSeq database (excluding *Paramecium* sequences) covered at least 20% of the scaffold length. If the mitochondrial genome (more or less fragmented) could be identified by a BLASTN (-evalue 1e-1 –perc_identity 70) against the *P. tetraurelia* mitochondrial genome, the scaffold (s) were tagged as mitochondrial. A handful of chimeric scaffolds were detected and corrected in the *P. octaurelia*, *P. primaurelia*, and *P. sexaurelia* assemblies by visual inspection of available long-range sequencing data (remapped mate-pairs) (see S2 Data).

## The constitutive MAC

Paired-end MAC DNA sequencing data were mapped on the MAC genome assembly using Bowtie2 (v.2.2.3 –local, otherwise default parameters) [72]. We defined the constitutive MAC as consisting of all regions of the assemblies with the expected average read coverage. We defined the regions of low coverage at scaffold extremities as MAC-variable regions. These regions result from the variability of programmed genome rearrangement patterns during MAC development [73]. While most MIC loci are either fully eliminated during MAC development (MIC-limited sequences) or fully retained (MAC-destined sequences), MAC-variable regions correspond to DNA sequences that are not completely eliminated and instead are retained in a small fraction of MAC copies. MAC-variable regions represent approximately 15% of the initial MAC genome assembly (S2 Table). In relation to the MAC DNA-seq depth,

a minimum expected coverage (v1.9 of samtools depth–q 10 –Q 10) was defined for each assembly (*P. octaurelia* 50X, *P. pentaurelia* 35X, *P. primaurelia* 20X, and *P. sonneborni* 35X). For each scaffold extremity, a Perl script analyzed the coverage in sliding 2 kb windows. The first window from the end of the scaffold with a coverage above the minimum expected coverage delimited the end of the MAC-variable regions. Only regions of minimum size 4 kb were kept. The script adjusted region ends using the MAC telomerisation sites and the ends of coding genes. After this automatic pipeline, each scaffold and mask was adjusted by eye using Circos drawings [74] (see example S6 Fig, representing DNA and RNA coverage, density in noncoding genes, and positions of the MAC telomerisation sites). The positions of the regions used to reconstruct the constitutive MAC for each MAC assembly are provided in S2 Data.

### IES annotation

Annotation of IESs was performed using the ParTIES toolkit [21] with default parameters. Briefly, this involves (i) alignment of MIC paired-end reads with a reference MAC genome to establish a catalog of potential IES insertion sites and to exclude reads that match perfectly across these sites hence do not contain IESs; (ii) assembly of the remaining reads with Velvet to obtain contigs that may contain IESs; and (iii) alignment of the contigs with the MAC reference genome to determine the position and the sequence of the IESs.

### Gene annotation

Gene annotation for the 9 species was carried out using a pipeline specifically tuned for the high gene density and tiny intron size (20 to 30 nt) characteristic of *Paramecium* somatic genomes. RNA-Seq transcriptome data were used to predict transcription units with the TrUC v1.0 software (https://github.com/oarnaiz/TrUC), as detailed in [36]. EuGene v4.1 software [75] configured with curated *Paramecium tetraurelia* genes [36] was used for ab initio predictions and to combine annotation evidence (the transcription units, the ab initio predictions, and comparative genomics evidence). Gene annotation completeness was calculated using BUSCO (v4.1.4—mode prot—lineage_dataset alveolate; [76]) through the Galaxy Portal (https://usegalaxy.org/).

### Assembly-free genome size estimation

Illumina paired-end sequencing reads were used to estimate genome size based on counting all substrings of 17 nt in the reads, using jellyfish software version 2.2.10 [77]:

```
jellyfish count -t 12 -C -m 17 -s 5G -o <sample.jf> <sample_paire-
d_end_reads.fastq>
jellyfish histo -o <sample.histo> <sample.jf>
```

The method for genome size estimation, described in [18,19], assumes that the total number of $k$-mers (in this case 17-mers) divided by the sequencing depth is a good approximation for genome size.

As discussed in [19] for *Paramecium* genomes, the histogram of $k$-mer depth for a perfect, homozygous genome with no repeated sequences (and no sequencing errors) is fit by a Poisson distribution, the peak corresponding to sequencing depth. The estimate of genome size is obtained by dividing the total $k$-mer count (excluding the peak near the origin that results from $k$-mers with sequencing errors) by the sequencing depth. This is straightforward for MAC genomes. As shown in S2 Table, the estimated MAC genome sizes were in good agreement with the size of the constitutive MAC genome assemblies.

For MIC genomes, variable amounts of contamination from MAC DNA lead to a second peak at higher $k$-mer depth corresponding to the sum of MAC-destined k-mers in the MIC

DNA and the MAC $k$-mers in the contaminating MAC DNA. This was only a significant problem for the *P. tetraurelia*, *P. sexaurelia*, and *P. sonneborni* MIC DNA samples, which were approximately corrected by assuming that a proportion of the $k$-mers counted from this second peak up to a depth of 500 were contributed by the contaminating MAC reads, while all the $k$-mers with a depth greater than 500, corresponding to highly repeated sequences, are of MIC origin (S2 Fig). The proportion of contaminating MAC DNA needed for this calculation was confirmed using IES retention scores (IRS) calculated with the MIC sequencing reads [21]. The position of the peak in the IRS distribution indicates the proportion of MIC (IRS approximately 1) and MAC (IRS approximately 0) DNA in the sample, as illustrated in S2 Fig.

## Identification of gene families

We performed an all-against-all BLASTP (ncbi-blast+ v. 2.2.30+) [78] search using the predicted protein sequences from each genome including also the proteins of *Tetrahymena thermophila* (June 2014 assembly http://ciliate.org) as an outgroup. From the resulting output, we determined gene families with SiLiX v. 1.2.9 [79]. The resulting gene families were aligned with MAFFT v7.305b (2016/Aug/16) [80] using the—auto option. Gene families with fewer than 3 genes or average pairwise identity less than 50% were excluded from downstream analyses. From the protein alignments, we reconstructed the nucleotide coding sequence alignments.

## Paramecium species phylogeny

To reconstruct the species phylogeny, we first selected single-copy gene families present in all 9 *Paramecium* species ($N = 1,061$ genes). When available, the *T. thermophila* homolog was also included as an outgroup. We estimated the maximum likelihood phylogeny using IQtree v.1.4.2 [81], considering each gene as a separate partition. We performed model testing on each partition and chose the best codon model (determined by the largest BIC). We evaluated the results by 1,000 bootstrap replicates. All internal branches but one are supported by 100% bootstrap values (Fig 1). We will hereafter refer to this species tree inferred from single-copy gene families as *Tree1*.

The rationale for analyzing single-copy gene families is that these sets of homologous sequences are a priori expected to correspond to orthologs. However, given that *Paramecium* have been subject to 3 rounds of whole-genome duplications followed by massive gene losses [82], it is possible that some single-copy gene families include paralogs. To check whether hidden paralogs might have biased the estimation of the species tree, we used PHYLDOG v.2.0beta (build 10/10/2016), a maximum likelihood method to jointly infer rooted species and gene trees, accounting for gene duplications and losses [25]. The analysis was performed using all gene families ($N = 13,617$). The default program options were used with additionally setting a random starting species tree and BIONJ starting gene trees. The duplication and loss parameters were optimized with the average, then branchwise, option, and the genomes were not assumed to have the same number of genes. We also ran PHYLDOG considering *Tree1* as the fixed species tree and keeping the remaining options identical. The topology of the most likely species tree inferred with PHYLDOG is almost identical to *Tree1* (it only slightly differs in the positions of *P. biaurelia* and *P. tredecaurelia*), and its likelihood is not significantly different from that obtained when running PHYLDOG with *Tree1* as a species tree. Thus, the species tree inferred by PHYLDOG using all gene families ($N = 13,617$) shows no significant disagreement with the phylogeny based on single-copy gene families (*Tree1*). We therefore hereafter considered *Tree1* as the reference species tree for all our analyses. To identify duplication and

speciation nodes in gene phylogenies, we computed reconciled trees for each gene family with PHYLDOG, using *Tree1* as a species tree.

## Taking into account the uncertainty of IES presence due to limited detection sensitivity

To identify events of IES gain and loss along the species phylogeny, it is necessary to analyze the pattern of presence/absence of IESs at homologous loci across species. One difficulty is that some IESs may remain undetected (false negatives). In particular, the sensitivity of Par-TIEs depends on the local read coverage [21]. To take into account the uncertainty arising from the variable local read coverage along scaffolds of each species, we calculated the coverage of MIC reads mapped against the MAC genome. We identified genes with extreme values of coverage (less than the 10th percentile or more than the 90th overall genes) or with an absolute read coverage of less than 15 reads. These genes correspond to regions with possible assembly errors or to regions of low power to detect IESs, and we marked them as problematic for IES annotation. IESs in these genes were considered to have an uncertain status of presence, and if no IES was annotated, the genes were marked as potentially containing IESs. To avoid issues due to genome assembly errors, we excluded from our analyses all IESs identified on small scaffolds (<10 kb).

## Taking into account the uncertainty of IES location (floating IESs)

To identify co-orthologous IES loci, i.e., that result from a single ancestral insertion event, we searched for IESs located at a same site across homologous sequences. It should be noted that the location of IESs, inferred from the comparison of MIC and MAC sequences, is sometimes ambiguous. This occurs when the IES boundaries overlap a motif repeated in tandem (S13 Fig). Such cases, hereafter called "floating IESs," represent 7% of all IESs. In the vast majority of cases (86%), the alternative locations of floating IESs differ by only 2 bp (as in the example shown in S13 Fig), and there are less than 1% of floating IESs for which the uncertainty in IES position exceeds 5 bp. To determine the exact location of IESs and capture the inherent ambiguity due to possible floating IESs, we used a 10-bp window around each annotated IES location to determine if the IES was classified as floating. If so, the alternative locations were added to the IES annotation.

## Co-orthologous IES insertion sites

To detect co-orthologous IES loci, we compared the position of IESs within homologous genes. To do so, we analyzed gene families with more than 3 sequences and average pairwise identity (at the protein sequence level) of more than 50%. To avoid ambiguity in the identification of homologous sites, we filtered protein sequence alignments with GBlocks v0.91b [83], and we only retained IESs located within conserved alignment blocks. An IES insertion site spans 2 nucleotides (TA). In a multiple sequence alignment including gaps, an IES locus can be larger (e.g., T—A). Two IES loci were considered as homologous if they have at least 1 shared site within the alignment (taking into account all potential locations in the case of floating IESs). In the case of floating IESs overlapping the boundaries of conserved alignment blocks, the presence or absence of homologous IES loci in other sequences cannot be reliably inferred. We therefore only retained IESs for which all homologs (if any) are entirely located within the conserved alignment blocks (i.e., we discarded sets of co-orthologous IES loci that included some floating IESs for which some of the possible alternative positions were located outside of the conserved alignment blocks).

## Ancestral state reconstruction and inference of IES insertion and loss rates

To explore the dynamics of IES gain and loss, we used a Bayesian approach to reconstruct the ancestral states of presence and absence of IESs using revBayes 1.0.0 beta 3 (2015-10-02) [84]. We constructed binary character matrices (presence/absence) for each gene family containing at least 1 IES unambiguously located within a conserved alignment block (see above). We assumed a model of character evolution with 1 rate of gain and 1 rate of loss sampled from the same exponential distribution with parameter α and a hyperprior sampled from an exponential with parameter 1. We excluded from the analysis 5 gene families for which revBayes could not compute a starting probability due to very small numbers. We used PHYLDOG reconciled gene trees (see above) to fix gene tree topologies and branch lengths. We ran $5 \times 10^5$ iterations. The search parameters were optimized in an initial phase of 10,000 iterations with tuning interval 1,000. Good sampling of the parameter space was verified by inspecting the time series and autocorrelation plots of the parameters. The convergence was validated by inspecting the multivariate Gelman and Rubin's diagnostic plots for different iterations.

Thus, for a given IES locus in a given gene family, revBayes provides an estimate of the probability of presence of an IES at each node of the gene phylogeny. We used these probabilities of presence along the gene phylogeny to estimate rates of IES gains or losses in each branch of the species tree. Because of gene duplications, a given branch in the species tree can be represented by several paths in the gene tree. Thus, we considered all paths in the gene tree that connect the corresponding speciation nodes (see S8 Fig for a simplified example). To measure the IES gain rate at a given IES locus ($c$), in a given gene family ($g$), we define $p^+_{cgij}$ as the sum of increase in probability of presence of an IES at this locus along all paths of gene family $g$ connecting speciation nodes $i$ and $j$ (where $i$ is a direct ancestor of $j$). Let $ng$ be the length in kilobase pairs of gene family $g$ alignment (counting only well aligned sites, where the presence of IESs can be assessed). Let $Ig$ be the number of IES loci in family $g$. Let $kgij$ be the number of paths connecting speciation nodes $i$ and $j$ in family $g$. Let $bij$ be the branch length connecting nodes $i$ and $j$ in the species tree ($bij$ is taken here as a proxy for time). Let $p^+_{gij}$ be the sum of increase in probability of presence of an IES, cumulated over all IES loci in family $g$. We define $p^+_{ij}$ as the sum of increase in probability of presence of an IES, cumulated over all IES loci along the path $i$ to $j$ of family $g$. We define $Gij$ as the rate of IES gain over all gene families ($f$) along path $ij$ expressed in number of IES gains per kilobase pairs of alignment per unit of time:

$$G_{ij} = \frac{\sum_{g=1}^{g=f} p^+_{gij}}{\sum_{g=1}^{g=f} n_g k_{gij}.b_{ij}}$$

We define in a similar manner the rate of IES loss. For a given gene family $g$, let $p^-_{cgij}$ be the sum of decreases in probability of presence of an IES in IES locus $c$ along a lineage in gene family $g$ connecting speciation nodes $i$ and $j$. Let $Ig$ be the number of IES loci in family $g$. Let $p^-_{gij}$ be the sum of decrease in probability of presence of an IES, cumulated over all IES loci in family $g$. We define as $Lij$ the rate of IES loss over all gene families ($f$) along path $ij$ expressed in number of IES losses per IES, per unit of time.

$$L_{ij} = \frac{\sum_{g=1}^{g=f} p^-_{gij}}{\sum_{g=1}^{g=f} I_g.k_{gij}.b_{ij}}$$

## IES age of insertion

The age of first insertion for each group of co-orthologous IES locations is defined as the age of the most recent common ancestor of all nodes in which an ancestral IES was present with probability larger than 99%.

## Identification of homologous IES sequences and characterization of mobile IESs

To characterize families of homologous IES sequences, we first compared all IESs (from all species) against each other with *blastn* (ncbi blast+ v2.5.0 [78]):

```
blastn -evalue 1e-8 -query IES.fa -db IES -dust yes -task blastn
-max_target_seqs 10000
```

We retained all pairs of homologous IESs for which BLAST alignments encompass the first and last 20 nt of the query and subject sequences. This ensures that the detected sequence homology includes the boundaries of the IESs and is not merely due to the presence of repeated sequences inserted within a preexisting IES.

To identify potentially mobile IESs, we searched for homologous IES sequences present at different (nonhomologous) genomic loci. For this, we extracted 100 nt on each side of the IES location and compared all these flanking regions against each other with *blastn* (using the same parameters as above). Pairs of homologous IES sequences with strong hits in flanking regions (≥75% identity over 150 nt or more) were classified as "homologous IESs at homologous loci." The other pairs were classified as "candidate mobile IESs." We clustered each group based on sequence similarity using SiLiX [79] with default parameters.

We further analyzed all clusters of candidate mobile IESs having at least 10 sequences ($N =$ 57 clusters). For each cluster, we constructed multiple sequence alignments with MAFFT v7.305b (with—adjustdirection and—auto options). We manually inspected these alignments to select full-length copies and create a multiple alignment covering the entire repeated element. At this stage, we excluded 11 clusters corresponding to very AT-rich sequences, for which it was not clear whether the detected sequence similarities were due to homology or to their highly biased sequence composition. Furthermore, 2 clusters were split into subfamilies, to include only sequences that are homologous over their entire length. We then used these seed alignments to build an HMM profile for each repeat family and search for homologous copies among the entire IES dataset with NHMMER version 3.1b2 [85].

In total, NHMMER identified 12,184 IESs having a significant hit (E-value $< 10^{-3}$) in the dataset of HMM profiles. Among detected hits, we distinguished 2 categories: (1) cases where the detected repeated element is located within the IES but does not overlap with the extremities of the IES (i.e., nested repeats); and (2) cases where the extremities of the HMM profile align with the extremities of the IES (with a tolerance of 3 bp to allow alignment uncertainties). IESs belonging to this latter category were hereafter considered as "mobile IESs." For subsequent analyses, we selected all families with more than 10 mobile IESs in at least 1 genome ($N =$ 24 families of mobile IESs). Multiple alignments, HMM profiles, and the list of matching IESs are available (https://doi.org/10.5281/zenodo.4836464).

## Supporting information

**S1 Fig. Multigate flow cytometry strategy for sorting the MICs.** GFP, DAPI-positive MICs from *P. sonneborni* vegetative cells transformed with the *P. tetraurelia CENH3a-GFP* transgene [62] were sorted based on size, granularity, DAPI staining, and GFP signal (see Materials and methods). P4 and P8 were sorted separately. Based on quality control by flow imaging (Imagestream) indicating 97% purity, the 2 samples P4 and P8, which represent 1.91% of total events,

were combined for DNA extraction and sequencing. Two populations are visualized and likely correspond to 2n and 4n MICs. FSC, forward scatter; GFP, green fluorescent protein; MIC, micronucleus; SSC, side scatter.
(TIF)

**S2 Fig. Estimating the proportion of MIC and MAC DNA in the sample based on IRS.** The histograms on the left show the k-mer depth profiles. The peak at the origin can be attributed to sequencing errors (k-mers that occur only once or a few times). The position of the largest peak beyond the origin corresponds to k-mers present once in the genome and provides the sequencing depth. As *P. aurelia* genomes have undergone whole genome duplications, there are a significant number of k-mers at 2X and even 4X the sequencing depth arising from genes (or regions of genes) present in 2 or 4 copies, clearly visible for *P. octaurelia* and *P. primaurelia*. The profile for *P. tetraurelia*, however, has a first peak (MIC sequences that occur once) at 31X followed by a larger peak that is not at the 2X position as it arises because of MAC DNA contamination. The column on the right shows histograms of IRS. Only the *P. tetraurelia* sample is significantly contaminated by MAC DNA: The average IRS of 0.4 indicates 40% MIC and 60% MAC DNA in this sample. The data underlying this figure may be found at https://doi.org/10.5281/zenodo.4836464. IES, internal eliminated sequence; IRS, IES retention score; MAC, macronucleus; MIC, micronucleus.
(TIF)

**S3 Fig. Comparison of cytometry and k-mer MIC genome size estimates.** Flow cytometry estimates of DNA content of micronuclei and k-mer counting estimates of genome size are described in Materials and methods. In order to show all of the data, both axes of the graph are log transformed. Simple linear regression was carried out on the untransformed data with R. The linear model that fits the data is presented as a dashed blue line; $R^2 = 0.99$, $p$-value $= 1.3 \times 10^{-09}$. The data underlying this figure may be found at https://doi.org/10.5281/zenodo.4836464. MIC, micronucleus; pbi, *P. biaurelia*; pca, *P. caudatum*; poc, *P. octaurelia*; ppe, *P. pentaurelia*; ppr, *P. primaurelia*; pso, *P. sonneborni*; pse, *P. sexaurelia*; pte, *P. tetraurelia*; ptr, *P. tredecaurelia*.
(TIF)

**S4 Fig. Repeat content of *P. caudatum* MIC genome.** (A) Abundance of repeat families identified by DNAPipeTE in *P. caudatum* strain My43c3d. The repeat content of the *P. caudatum* MIC genome was analyzed with DNAPipeTE [86], using a sample of 3,500,000 sequence reads (corresponding to a read depth of approximately 0.5X). DNAPipeTE identified 67 repeat families that collectively constitute 83% of the MIC genome. Among them, there are 2 major satellite repeats Sat1 and Sat2, which represent, respectively, 42% and 29% of the MIC genome. The data underlying this panel may be found at https://doi.org/10.5281/zenodo.4836464. (B) Sequences of the 2 major satellite repeats Sat1 and Sat2 in *P. caudataum* My43c3d (332 bp and 449 bp long). These 2 satellite repeats share homology over an approximately 200-bp-long region. Primer sequences used for specific PCR amplification of each repeat are indicated in bold. (C) Detection of Sat1 and Sat2 *in P. caudatum* strains. Whole-cell genomic DNA was used to perform duplex PCR with a set of primers located within each repeat (Sat1 or Sat2, in bold panel B) and another set of primers within the 18S ribosomal DNA as a loading control. The expected size of the 18SrDNA PCR product was 301 bp using primers 18S_F953: AGAC GATCAGATACCGTCGTAG and 18S_R1300: CACCAACTAAGAACGGCCATGC. L: 1-kb NEB ladder. Neg.: negative control (no DNA). Sat1 was amplified with primers comp2975_F1: TTGTGCTGTAGGGCTCAATAAT and comp2975_R1: CTCAAAATTCGACGCTGACAA at the expected size (198 bp) in the *P. caudatum* clade B strains tested (My43c3d; C033; C083; C131; C147). The repeat could not be amplified in *P. caudatum* DNA from clade A strains

(C023; C065; C104; C119), from strain C026 or from strain Indo_1.6I. Sat2 was amplified with primers comp5240_F1: TGCTGCTGATTTTGGATCTCG and comp5240_R1: CCGAGAAC GGCCATTACAAG at the expected size (168 bp) in the *P. caudatum* clade B strains tested (My43c3d; C033; C083; C131; C147). The repeat could not be amplified in *P. caudatum* DNA from clade A strains (C023; C065; C104; C119), from strain C026 or from strain Indo_1.6I. MIC, micronucleus.
(TIF)

**S5 Fig. Impact of sequencing depth on IES detection.** To assess the impact of sequencing depth on the sensitivity of IES detection, we subsampled sequence reads from the *P. tetraurelia* MIC dataset so as to obtain subsets of lower depth (from 2× to 30×), on which we applied the same IES detection procedure. (A) Number of detected IESs vs. sequencing depth. (B–H) Length distributions of IESs detected within each subset (the percentage of IESs in each peak is indicated). Sequencing depth affects the number of detected IESs (for depths <15×), but not their length distribution. The data underlying this figure may be found at https://doi.org/10. 5281/zenodo.4836464. IES, internal eliminated sequence; MIC, micronucleus.
(TIF)

**S6 Fig. Example of an MAC-variable region.** Circular representation of 1 scaffold of approximately 730 kb. The tracks from the exterior to the interior of the circle: G+C content of 100 nt sliding windows (black), MAC DNA-seq depth (purple), the density in predicted noncoding genes (orange), RNA-Seq depth (red), and the density of detected telomerisation sites (green). The external blue arc shows the region identified as being MAC variable. These regions were determined by an automatic pipeline (see Materials and methods), then adjusted by eye for each scaffold. MAC, macronucleus.
(TIF)

**S7 Fig. Intragenic IES density vs. gene expression level.** Expression levels (RPKM) were measured with RNA-Seq datasets from vegetative cells. For each species, expressed genes were classified into 10 bins of equal sample size according to their expression level, and IES density computed within each bin. Nonexpressed genes (6.6% of the entire dataset) were excluded. (A) *P. aurelia* species. (B) *P. caudatum*. The data underlying this figure may be found at https:// doi.org/10.5281/zenodo.4836464. IES, internal eliminated sequence; pbi, *P. biaurelia*; pca, *P. caudatum*; poc, *P. octaurelia*; ppe, *P. pentaurelia*; ppr, *P. primaurelia*; pso, *P. sonneborni*; pse, *P. sexaurelia*; pte, *P. tetraurelia*; ptr, *P. tredecaurelia*.
(TIF)

**S8 Fig. Measuring the rate of IES gain or loss along the species phylogeny.** To illustrate our methodology, we show here an example of a gene family with 3 genes, 2 from *P. sonneborni* (pson1, pson2) and 1 from *P. sexaurelia* (psex1). Two IES loci are found in this family (A, B). The probability of presence of an IES (estimated by Bayesian ancestral state reconstruction— see Materials and methods) is indicated by shaded circles for each locus at each node of the gene phylogeny. We focus here on the branch of the species tree leading from the common ancestor of *P. sexaurelia* and *P. sonneborni* to the leaf node of *P. sonneborni* (the red branch in the species tree, shown in the insert). The length of this branch ($b$) is taken as a proxy for time. Because of a duplication event, this branch of the species tree corresponds to 2 paths in the gene tree ($k = 2$). To estimate the IES gain rate, we calculate for each path the sum of increase in the probability of presence of an IES, for all IES loci ($p^+$). Along the first path (from the root to pson1), we have $p^+A1 = 0.5$ and $p^+B1 = 0$. Along the second path (from the root to pson2), we have $p^+A2 = 0.5$ and $p^+B2 = 0$. The average gain rate along all paths, per unit of time and per bp, is thus given by $G = (p^+A1 + p^+B1 + p^+A2 + p^+B2) / (k \times b \times n_g)$, where $n_g$ is the number

of well-aligned sites in the gene family alignment (i.e., the number of sites where the presence of co-orthologous IESs can be assessed). Similarly, to estimate the IES loss rate, we calculate for each path the sum of decrease in the probability of presence of an IES, for all IES loci ($p^-$). Along the first path (from the root to pson1), we have $p^-A1 = 0$ and $p^-B1 = 0.4$. Along the second path (from the root to pson2), we have $p^-A2 = 0$ and $p^-B2 = 0.4$. The average gain rate along all paths, per unit of time and per bp, is thus given by $L = (p^-A1 + p^-B1 + p^-A2 + p^-B2) / (k \times b \times I)$, where $I$ is the number of IES loci in the gene family (here $I = 2$). IES, internal eliminated sequence.
(TIF)

**S9 Fig. Dating events of IES insertion/loss.** (A) To date events of IES loss or gain, it is first necessary to identify IESs that derive from a single ancestral insertion event (co-orthologous IESs). For this, we aligned coding sequences (based on the protein alignment) and mapped the position of IESs: IESs located at the exact same position within a codon were assumed to be co-orthologous. We then used the reconciled gene tree to map events in the species phylogeny, using a maximum likelihood approach (see Materials and methods). The example shown here corresponds to a gene family encoding a putative RNA 3′-terminal phosphate cyclase (PTET.51.1.P0920097, POCTA.138.1.P0960088, PBIA.V1_4.1.P01950012, PTRED.209.2.P71800001293600070, PPENT.87.1.P1090087, PPRIM.AZ9–3.1.P0020612, PSON.ATCC_30995.1.P0860097, PSEX.AZ8_4.1.P0910047, PCAU.43c3d.1.P00760109). The positions of IESs are indicated by red rectangles. (B) The presence of IESs (red bars) within each of these genes is indicated with regard to the species phylogeny. Six distinct IESs were identified in this gene family: IES2 is shared by all species and therefore predates the divergence between *P. caudatum* and the *aurelia* clade; IES4 most probably corresponds to a gain in the *P. sexaurelia* lineage; IES5 and IES6 predate the divergence of the *aurelia* clade and have been subsequently lost in the *P. tetraurelia/P. octaurelia* lineage; IES1 might correspond to a gain at the base of the *aurelia* clade or a loss in the *P. caudatum* lineage (and vice versa for IES3). IES, internal eliminated sequence.
(TIF)

**S10 Fig. Prevalence of weak IESs.** (A) Proportion of weak IESs (i.e., IESs with a retention frequency ≥10% in WT vegetative cells) among IESs located in different genomic compartments. The proportion of weak IESs among intergenic IESs was compared to that of IESs located in introns or coding regions, by a chi-squared test. (B) Proportion of weak IESs according to the age of IESs (for IESs located in coding regions): New = species-specific IES; Old = IES predating the divergence between *P. caudatum* and the *aurelia* lineage. The number of new IESs is indicated for each species. The proportion of weak IESs among new IESs was compared to that of older ones, by a chi-squared test. Species codes: pbi, *P. biaurelia*; pca, *P. caudatum*; poc, *P. octaurelia*; ppe, *P. pentaurelia*; ppr, *P. primaurelia*; pso, *P. sonneborni*; pse, *P. sexaurelia*; pte, *P. tetraurelia*; ptr, *P. tredecaurelia*. (*: $p$-value < 0.05; **: $p$-value < 1e-3; ***: $p$-value < 1e-6). The data underlying this figure may be found at https://doi.org/10.5281/zenodo.4836464. IES, internal eliminated sequence; WT, wild-type.
(TIF)

**S11 Fig. Length distribution of IESs according to their age.** Comparison of the length distribution of IESs according to their age (for the subset of datable IESs located in coding regions). The age of an IES site is defined as in Fig 3. Results for other species are shown in Fig 4. The data underlying this figure may be found at https://doi.org/10.5281/zenodo.4836464. IES, internal eliminated sequence; pca, *P. caudatum*; ppe, *P. pentaurelia*; ppr, *P. primaurelia*; pso, *P.*

*sonneborni*; pse, *P. sexaurelia*; ptr, *P. tredecaurelia*.
(TIF)

**S12 Fig. Genomic distribution of IESs according to their length.** Green bars indicate the percentage of IESs located in each compartment of the MAC genome (introns, protein-coding regions, and intergenic regions) for each species. Gray bars indicate the percentage of the MAC genome in each compartment. (A) Long IESs (>100 bp). (B) Short IESs (<35 bp). For each species, the relative proportion of IESs in the 3 compartments (intron, intergenic, and coding regions) was compared for short IESs (<35 bp) vs long IESs (>100 bp) by a chi-squared test (*p*-value $< 10^{-16}$ in all species). The data underlying this figure may be found at https://doi.org/10.5281/zenodo.4836464. IES, internal eliminated sequence; MAC, macronucleus.
(TIF)

**S13 Fig. Example of floating IES.** Comparison of MIC and MAC sequences indicates the presence of an IES at this locus. However, because of the presence of a repeated motif at the boundaries of the IES (blue arrows), it is not possible to determine which of the 2 possible segments (IES-1 in black or IES-2 in red) is actually excised in vivo. Such IESs that cannot be unambiguously positioned are called "floating IESs." They represent 6.8% of the 400,254 IESs detected across all species. In the vast majority of cases (86%), the alternative locations of floating IESs differ by only 2 bp (as in the example shown here), and there are less than 1% of floating IESs for which the uncertainty in IES position exceeds 5 bp. IES, internal eliminated sequence; MAC, macronucleus; MIC, micronucleus.
(TIF)

**S1 Table. MIC genomes: Sequencing data and size estimates.** (a) Highly enriched sorted MIC (97%–99%) are contaminated with the highly polyploid MAC. For example, 97% of MIC purity led to 60% of MAC DNA contamination in the case of *P. tetraurelia* (see S2 Fig). MIC genome size (in Mb) was estimated based on (b) flow cytometry analysis and (c) k-mer counts. Size estimation before correction, based on MAC contamination, is indicated in parentheses (see Materials and methods). MAC, macronucleus; MIC, micronucleus.
(XLSX)

**S2 Table. MAC genome assemblies used in this study.** The assemblies of the 4 MAC genomes sequenced in the course of this project include both "constitutive MAC" regions (i.e., regions that are always retained in the MAC) and "MAC-variable regions" (i.e., regions that are mostly restricted to the MIC but that are retained at low frequency in MAC nuclei). The size and content of these 2 types of regions are indicated. The completeness of MAC genome assemblies was assessed using BUSCO [76]. MAC, macronucleus; MIC, micronucleus.
(XLSX)

**S3 Table. Distribution of IESs in different genomic compartments.** Values in parentheses indicate the proportions expected under the hypothesis of uniform IES distribution along MAC-destined regions. IES, internal eliminated sequence; MAC, macronucleus.
(XLSX)

**S4 Table. Genomic distribution of copies of mobile IESs.** Copies of mobile IESs were searched within IES sequences (see Materials and methods). Detected copies are divided in 2 categories: nested copies (i.e., copies inserted within an IES, but not including the extremities of the IES) and bona fide mobile IESs (i.e., copies whose extremities correspond to the extremities of the IES). This table lists all families for which at least 1 species contains ≥10 copies of bona fide mobile IESs in its genome. The genomic distribution (in coding regions, introns,

and intergenic regions) of IESs containing these copies (nested copies or bona fide mobile IESs) is indicated. IES, internal eliminated sequence.
(XLSX)

**S5 Table. Highly conserved IESs that are transcribed during MAC development and/or associated to genes that are up-regulated during MAC development.** The transcription level of IESs is indicated for different stages during MAC development (S, T0 to T45) and in vegetative cells (V) [36]. Only *P. tetraurelia* IESs are shown in the table, because this is the only species for which developmental transcriptome data are available [36]. (*) The IES pte. MICA.16.324097 (FAM_4968) contains a complete protein-coding gene, which is expressed during development (HTH CenpB-type DNA-binding domain; see Fig 5A). The gene in which this IES is inserted (PTET.51.1.P0160202) is not specifically expressed during development. IES, internal eliminated sequence; MAC, macronucleus.
(XLSX)

**S6 Table. Genetic distances between species of the *aurelia* complex.** This matrix reports for each pair of species, the number of orthologous genes analyzed (above the diagonal), and their median synonymous divergence (below the diagonal).
(XLSX)

**S7 Table. Sequencing data generated for this study.**
(XLSX)

**S1 Data. Flow cytometry-based estimations of MIC genome size in *Paramecium*.** MIC, micronucleus.
(XLSX)

**S2 Data. Locations of MAC-variable regions and MAC assembly curation.** This file provides the positions of MAC-variable regions identified in the MAC assemblies of *P. octaurelia*, *P. pentaurelia*, *P. primaurelia*, and *P. sonneborni*. In addition, it indicates the positions of putative assembly chimeras that have been identified in *P. octaurelia*, *P. primaurelia*, and *P. sexaurelia*. MAC, macronucleus.
(XLSX)

## Acknowledgments

We thank Damien de Vienne for his help in phylogenetic analyses and Alexey Potekhin (Saint Petersburg State University, St Petersburg, Russia) and Ewa Przybos (Institute of Systematics and Evolution of Animals, Polish Academy of Sciences, Cracow, Poland) for the maintenance of *Paramecium* stock collections and providing some *Paramecium aurelia* strains. This work was performed using the computing facilities of the CC LBBE/PRABI.

## Author Contributions

**Conceptualization:** Diamantis Sellis, Frédéric Guérin, Olivier Arnaiz, Karine Labadie, Eric Meyer, Linda Sperling, Laurent Duret, Sandra Duharcourt.

**Data curation:** Diamantis Sellis, Olivier Arnaiz, Linda Sperling, Laurent Duret, Sandra Duharcourt.

**Formal analysis:** Diamantis Sellis, Frédéric Guérin, Olivier Arnaiz, Emmanuelle Lerat, Linda Sperling, Laurent Duret, Sandra Duharcourt.

**Funding acquisition:** Eric Meyer, Laurent Duret, Sandra Duharcourt.

**Investigation:** Diamantis Sellis, Frédéric Guérin, Olivier Arnaiz, Emmanuelle Lerat, Nicole Boggetto, Sascha Krenek, Arnaud Couloux, Jean-Marc Aury, Karine Labadie, Sophie Malinsky, Simran Bhullar, Eric Meyer, Linda Sperling, Laurent Duret, Sandra Duharcourt.

**Methodology:** Diamantis Sellis, Frédéric Guérin, Olivier Arnaiz, Walker Pett, Nicole Boggetto, Arnaud Couloux, Jean-Marc Aury, Karine Labadie, Linda Sperling, Laurent Duret, Sandra Duharcourt.

**Project administration:** Karine Labadie, Sandra Duharcourt.

**Resources:** Frédéric Guérin, Olivier Arnaiz, Nicole Boggetto, Sascha Krenek, Thomas Berendonk, Arnaud Couloux, Jean-Marc Aury, Karine Labadie, Sophie Malinsky, Simran Bhullar, Linda Sperling, Laurent Duret, Sandra Duharcourt.

**Software:** Olivier Arnaiz, Arnaud Couloux, Jean-Marc Aury, Linda Sperling, Laurent Duret.

**Supervision:** Karine Labadie, Eric Meyer, Linda Sperling, Laurent Duret, Sandra Duharcourt.

**Validation:** Diamantis Sellis, Frédéric Guérin, Olivier Arnaiz, Sascha Krenek, Eric Meyer, Linda Sperling, Laurent Duret, Sandra Duharcourt.

**Visualization:** Diamantis Sellis, Frédéric Guérin, Olivier Arnaiz, Emmanuelle Lerat, Sascha Krenek, Eric Meyer, Linda Sperling, Laurent Duret, Sandra Duharcourt.

**Writing – original draft:** Diamantis Sellis, Frédéric Guérin, Eric Meyer, Linda Sperling, Laurent Duret, Sandra Duharcourt.

**Writing – review & editing:** Olivier Arnaiz, Sascha Krenek, Sophie Malinsky, Simran Bhullar, Eric Meyer, Linda Sperling, Laurent Duret, Sandra Duharcourt.

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
