## [Editor Report · Decision Letter 0]

20 Jan 2021

Dear Dr Duharcourt, 

Thank you for submitting your manuscript entitled "Massive colonization of protein-coding exons by selfish genetic elements in Paramecium germline genomes" for consideration as a Research Article by PLOS Biology.

Your manuscript has now been evaluated by the PLOS Biology editorial staff, as well as by an academic editor with relevant expertise, and I'm writing to let you know that we would like to send your submission out for external peer review.

Please re-submit your manuscript within two working days, i.e. by Jan 22 2021 11:59PM.

Kind regards,

Roli Roberts

Senior Editor

PLOS Biology

---

## [Decision Letter · Decision Letter 1]

3 Mar 2021

Dear Dr Duharcourt,

Thank you very much for submitting your manuscript "Massive colonization of protein-coding exons by selfish genetic elements in Paramecium germline genomes" for consideration as a Research Article at PLOS Biology. Your manuscript has been evaluated by the PLOS Biology editors, an Academic Editor with relevant expertise, and by three independent reviewers.

You'll see that the reviewers are broadly positive about your study, but each raises a number of concerns that will need to be addressed before further consideration. While most of these are textual and/or presentational, several of them will involve new analyses and possibly new data.

In light of the reviews (below), we are pleased to offer you the opportunity to address the comments from the reviewers in a revised version that we anticipate should not take you very long. We will then assess your revised manuscript and your response to the reviewers' comments and we may consult the reviewers again.

We expect to receive your revised manuscript within 1 month.

**IMPORTANT - SUBMITTING YOUR REVISION**

*Resubmission Checklist*

*Published Peer Review*

*PLOS Data Policy*

*Blot and Gel Data Policy*

Sincerely,

Roli Roberts

Senior Editor,

rroberts@plos.org,

PLOS Biology

REVIEWERS' COMMENTS:

Reviewer #1:

Using whole genome sequencing of the MIC and MAC genomes of nine Paramecium species, Sellis and Guérin present a convincing and highly significant demonstration of what has been suspected in the field for decades -- that the IES sequences in these species are born from transposable elements. It is hard to imagine any other way this elaborate and amazing state of the genome could arise, but the presumed solution has long lacked convincing empirical evidence. The key innovation here was the group's generation of whole genome sequences for diverse Paramecium species. In some cases the cross-species comparison of IES sequences may have been useful, but the limiting factor in previous attempts to solve this problem seems to have been that the model genome (P. tetraurelia) has few IESs with detectable repeat character, compared to some species with many more. The current study overcomes this so show that most species have many IESs derived from repeats and repeats that are likely transposable elements. The rest of the paper analyzes several fascinating aspects of these genome sequence including a presumed horizontal transfer of TEs and co-option of some TE/IES sequences for host genome function. The major flaws in the paper are in the strength of the stated conclusions of the evolutionary analyses and some unclear methodological details, which can be fixed with text changes. Overall, this paper will be an important addition to the ciliate MIC/MAC literature and of interest to a broad audience in genomic and evolution.

Major points

 1) The horizontal transfer model needs significantly expanded explanation. Are the authors suggesting the horizontal transfers include large segments of DNA or they are ISE/TE-specific. Judging by the shared clade between Pso and Ptr on Figure 2B, the shared ISE family is likely a byproducts of larger scale horizontal transfer between the two species. Given the availability of synteny information based on genome sequencing data, an analysis on whether the horizontal transfer included the flanking regions of FAM_2183 could clarify. The highly interspersed pattern of each species' TEs in figure 2 is mechanistically confusing. The base expectation of one or a few TEs jumping species would be a very different tree.

 2) The authors should define and clarify their analysis for concluding that there is high sequence conservation and therefore ("beneficial for Paramecium") amongst the member of a repeat family amongst different and distant species. This is not reasonable to conclude so broadly. What defines "highly conserved?" How are we to interpret conservation of a family of repeats? In many other species (and here for fig 5C), specific insertions have been shown to be co-opted, but why or how would a whole family be co-opted? This seems unlikely since most of the insertions should not be beneficial. Alternatively, the authors present some clear and convincing examples of suspected co-option in Fig 5. Without overstating the interpretation of the "conservation" in the 69 subfamilies, the authors could present the data in fig5 as examples that seem likely to be functional, and then state that they also find some other sequences with signatures of conservation which it will be interesting to investigate whether they too could be functional.

 3) The authors frequently use language that puts undue and unnecessary confidence behind interpretation of evolutionary analyses or fail to mention alternative interpretations of their data. For example, the statement (not speculation) that "(line 381 and 572) after their insertion, IESs progressively accumulate changes that make them more efficiently excised" is not supported by the data. Are there sequence motifs in these older IESs that correspond to the higher fidelity excision? Alternatively, the host genome has had more time to adapt to these sequences or a higher fidelity excision mechanism is used for older versus younger IESs. This model would parallel our understanding of KZnFs in mammals. Similarly, in lines 391-394, Ezl1 could be a young mechanism of restriction, older sequences are and have always been restricted with another gene/mechanism.

Minor points:

Line 86-91:

Why did the previous analysis of P. tetraurelia find so many fewer copies and families of repeats compared to this approach? Is the current analysis a much improved genome assembly?

Line 95 and discussion: 

It would be helpful to elaborate on the model of IESs originating from MAC sequences. As provided, the mechanism by which this could happen is not clear.

Line 122:

The details of the genome sequencing are dense and unnecessary in the text, especially in a broad readership journal like PLOS. Much of this could be moved to supplementary or methods to provide a more accessible and readable text. The first paragraph of the discussion could be used to cover most of the important details of the sequencing.

Table 1 also seems dense and unnecessarily detailed for the main text.

Table 2 is also dense with a lot of information that is not of primary relevance. I would suggest this table contains the key data from the sparse matrix and the rest moved to supplementary (which is a single arrow click away in PLOS). For example, the left half showing repeat-containing stats is secondary to the mobile IES data.

Lines 266-270 and other places throughout (lines 498-501):

The authors should take care with their use of the term 'homologous'. For example - "IESs that are homologous, i.e. that result from a single ancestral insertion event". While IES sequences that are found in syntenic loci in two genomes are most likely homologous, so are all the IESs at other loci in these genomes that share significant sequence similarity. It would be better to use orthologous or, as the authors use elsewhere, 'homologous IESs at homologous insertion sites', or invoke synteny.

Line 324:

"absence of homology" should be "absence of sequence similarity"

Line 339-340:

The text states that FAM_3 is as old as the LCA of subclade A, but then states that "all …IESs that we detected have been subject to recent waves of insertion?"

Line 348-350:

TEs are the exemplar homoplasy-free mutation. The statement that "these two copies correspond to independent insertion events at a same site, rather than ancestral events", needs considerable support. Is there a strong insertion bias of these elements that would limit the potential number of sites? Otherwise, this seems statistically highly unlikely given genome size and number of insertions of this family.

Line 384-386:

What was the rational for doing a single knockdowns of Dcl2/3, since the text states that genes require either Ezl alone or both Ezl and Dcl?

Lines 448-45:

"Among the 56 families of highly conserved IESs present in P. tetraurelia, 10 (18%) are transcribed at substantial levels (>1 RPKM) during autogamy (as compared to 0 0.8% for other IESs) (S4 Table)." Wouldn't we expect that less diverged, younger sequences would be more likely to retain promoter activity and therefore be transcribed, without need to invoke co-option?

Line 516:

"must correspond to non-autonomous elements" should be "likely correspond…"

Lines 590-596:

Are the genes required for excision near the insertion sites of these "conserved" families of IESs? This seems highly speculative and should be rephrased to reflect this.

Lines 719-720:

"the separation of somatic and germline functions between the MIC and the MAC offered the possibility for selfish genetic elements to invade genes in ciliates." Isn't it equally or more likely that the invasion of TEs drove the complex biology of separating germ/soma via MIC/MAC?

Reviewer #2:

This is a well-written and well-argued manuscript that focuses on the evolutionary dynamics of germline-limited sequences (i.e. IESs) in multiple species of the genus Paramecium. The authors use a comparative approach that combines high-throughput sequencing of isolated germline micronuclei (and occasionally somatic macronuclei) with a suite of bioinformatic tools/approaches. Most importantly, the analyses and resulting insights on IES dynamics are well-justified, and the manuscript is generally written in a manner that will make the complex biology of ciliates, plus the interest results here, accessible to broad PLOS Biology readership. 

MAJOR CONCERNS

- I am confused by the claim that thousands of IESs were acquired by horizontal transfer… are these really thousands of HGTs or a smaller number coupled with intragenomic expansions (lines 108, 542-545).

- There is perhaps a bit too much conclusion/discussion in the intro… and missing is a brief intro to the murkiness/history of species designations in the genus Paramecium. Similarly, parts of the discussion is a bit overwritten (see below) and perhaps some of the methods could be move to the supplement, though this would be an editorial decision.

- The final section of the discussion, on IESs and spliceosomal introns, seems overwritten given the data. For example, the claim that the "the nuclear envelope opened the way for introns to invade genes' requires both citation (see, for example, papers by William Martin and Eugene Koonin) and a more nuanced interpretation. I suggest the authors tighten substantially here to better reflect the data in this manuscript - at least some of this long speculative section might be better in a subsequent review paper. 

- The sequence logos and phylogenetic analyses (Fig 2) are not particularly compelling. For one, the first sequences seems to reflect just the compositional bias (i.e. AT richness) so here, reanalyzing using shuffled sequences AND reporting on these data is critical. Also, what is the length variation for Fam_2183… are the all 233bp with no variance? Similarly, the 'phylogeny' is suspect as rapid rates of evolution may make alignment/homology assessment very difficult here. In other words, the star-like shape could be due to either rapid burst OR (as discussed elsewhere in the manuscript) rapid rates of evolution obscuring homolog assessment. I would revisit effect of compositional bias (for motifs) and homology assessment (for tree), and then perhaps move this part to the supplement.

MINOR CONCERNS:

- Line 64: word choice… not really 'destroyed' but instead recycled?

- Measure of 'substantial burden' (line 110) might best be done with fecundity studies… also, there is a bit too much conclusion in the section of the intro. Please rewrite/rethink.

- Similarly, the section entitle 'fitness consequences' (line 555) might be better toned down/tightened… no fitness is measured here, just inferred. So perhaps 'Possible fitness consequences…' 

- A bit more discussion of MIC genome size and % of retention in MAC genomes might be warranted here, or in a subsequent manuscript. Similarly, more discussion of relationship between IES presence and gene expression data would also be nice. These are such impressive comparative data.

- The periodicity of IES lengths might be worth some additional attention.

- The logic of the section on "why IESs are not eliminated" (598…) feels a bit tautological/circular, and the too-long paragraph here makes this section a bit difficult to follow. Please rethink/refine, and perhaps consider the concept of evolutionary constraint. Perhaps some/many IESs are functionally neutral, which may be consistent with the observation of loss of IESs from highly expressed genes? I am suggesting rewording and shortening here.

- Table 2 is a bit difficult to interpret as presented… especially given the discussion of TEs that follows. I would encourage the authors to rethink there, perhaps synthesizing the data a bit more to explicitly link to TE families and moving this detailed version to supplemental materials

- From figure 1 legend: the claim that estimates of loss rate along terminal branches … may be overestimated is true, but doesn't the logic also go the other way - that some IESs may have been gained and lost, and therefore undetected…

- Figure 3 is very cool, but not accessible to general reader with current legend/titles.

- I wonder if there is a way to add an inset to Fig S14, or at least to point to it in legend, of Figure 1

Reviewer #3: 

Summary: 

In this report Sellis et al. studied multiple Paramecium species and examined the evolutionary history of IES gain throughout the clade. The authors identified IES sequences by assembling the macronuclear genomes of these species and examining inserted sequences from micronuclear reads when compared to the MAC assemblies. Once each species' repertoire of IESs had been identified, the authors inferred the relative age of IESs by examining the presence or absence of certain IESs between the different species in comparison with their phylogenetic distribution. The authors found that older IESs are more easily excised from the genome during rearrangement, requiring fewer excision pathways to be successfully removed. They discovered multiple families of IESs that show evidence of recent acquisition and expansion, with certain families having hundreds to thousands of copies present within different species, some of which share homology with known transposable elements. Their results provide evidence that IESs are derived from degenerated transposable elements, which has long been hypothesized to be the case in the field. In addition, they identify a set of IESs encoding a protein under strong purifying selection, suggesting that certain IESs have a beneficial effect on the fitness of these species through either encoding proteins or regulating gene expression. The work in this manuscript is sound, but there are some minor issues that should be addressed before it is published.

Specific comments:

1. The main text has an abundance of specific numbers scattered throughout that can already be found in the tables. Removing some of these numbers and referencing the tables in the text instead would improve the readability of some sections for a more general audience.

2. While the data presented in this report is generally robust, it suffers from a lack of statistical testing. Adding significance testing would strengthen some of the conclusions, such as in figures 3 or S9.

3. In the Results section on page 7, the portion of text describing the methods used for genome size estimation should be moved to Methods.

4. Since the sequencing depth of the MIC genomes has a significant impact on the identification of IESs, details of the MIC genome coverage should be included in Table 1.

5. The high proportion of repeat content in the P. caudatum MIC is curious. Could this be an artifact of overamplified reads in the sequencing library? Adding another method of repeat content estimation, such as some long-read Nanopore or PacBio sequencing, could help reinforce this claim, if such data are easily generated or already available.

6. Many MAC genome assemblies were assembled for this study, and the IES detection process is dependent on the quality of the assembled genomes. Some measurements of genome completeness, such as BUSCO scores or tRNA representation, would be a good addition, and are usually used to assess completeness of other ciliates MAC genomes..

7. It is claimed that the presence of certain families in both P. sonneborni and P. tredecaurelia is suggestive of horizontal transfer. Do the geographic distributions of these species overlap? Do they share any food sources or symbionts that could have been involved in this transfer? Can a scenario of shared ancestry followed multiple losses/degeneration in all the other lineages be entirely ruled out? Are some IES families not detected because of low MIC sequencing depth?

8. A scalebar would be nice for the tree on the right of Fig. 3.

9. For Figure 4A, the low coverage and high repeat content of the sequencing for P. caudatum makes me skeptical of its chart. Given the disparity in the amount of data between the clades, the comparison between them does not seem too meaningful. The chart of the aurelia clade alone would be fine. 

10. Figure S7B does not have a label on its Y-axis. Intergenic IESs will be harder to distinguish because they atrophy faster, so there will always be a bias towards annotating IESs in coding regions that interrupt MDSs, over intergenic background.

---

## [Editor Report · Decision Letter 2]

24 May 2021

Dear Dr Duharcourt,

Thank you for submitting your revised Research Article entitled "Massive colonization of protein-coding exons by selfish genetic elements in Paramecium germline genomes" for publication in PLOS Biology. The Academic Editor kindly agreed to check your revisions and responses to the authors, thereby sparing you a further round of review.

Based on the Academic Editor's favourable assessment, we will probably accept this manuscript for publication, provided you satisfactorily address my data and other policy-related requests.

IMPORTANT: Please attend to my Data Policy requests below. Essentially, we need you to provide the numerical values for Figs 1, 2AC, 3, 4AB, S2, S3, S4A, S5ABCDEFGH, S7AB, S10AB, S11, S12AB. It's unclear whether these are in your current Zenodo deposition; if so, please clarify. If not, please provide these, either as supplementary data files or as a distinct part of your Zenodo deposition. Either way, please cite the location of the data clearly in each relevant Figure legend (e.g. "The data underlying this Figure may be found in XYZ").

We expect to receive your revised manuscript within two weeks. 

*Published Peer Review History*

*Early Version*

Sincerely,

Roli Roberts

Senior Editor,

rroberts@plos.org,

PLOS Biology

DATA POLICY:

Regardless of the method selected, please ensure that you provide the individual numerical values that underlie the summary data displayed in the following figure panels as they are essential for readers to assess your analysis and to reproduce it: Figs 1, 2AC, 3, 4AB, S2, S3, S4A, S5ABCDEFGH, S7AB, S10AB, S11, S12AB. NOTE: the numerical data provided should include all replicates AND the way in which the plotted mean and errors were derived (it should not present only the mean/average values).

We require the original, uncropped and minimally adjusted images supporting all blot and gel results reported in an article's figures or Supporting Information files. We will require these files before a manuscript can be accepted so please prepare and upload them now. Please carefully read our guidelines for how to prepare and upload this data: https://journals.plos.org/plosbiology/s/figures#loc-blot-and-gel-reporting-requirements 

DATA NOT SHOWN?

---

## [Editor Report · Decision Letter 3]

4 Jun 2021

Dear Sandra,

On behalf of my colleagues and the Academic Editor, Harmit Malik, I'm pleased to say that we can in principle offer to publish your Research Article "Massive colonization of protein-coding exons by selfish genetic elements in Paramecium germline genomes" in PLOS Biology, provided you address any remaining formatting and reporting issues. These will be detailed in an email that will follow this letter and that you will usually receive within 2-3 business days, during which time no action is required from you. Please note that we will not be able to formally accept your manuscript and schedule it for publication until you have made the required changes.

PRESS: We frequently collaborate with press offices. If your institution or institutions have a press office, please notify them about your upcoming paper at this point, to enable them to help maximise its impact. If the press office is planning to promote your findings, we would be grateful if they could coordinate with biologypress@plos.org. If you have not yet opted out of the early version process, we ask that you notify us immediately of any press plans so that we may do so on your behalf.

Thank you again for supporting Open Access publishing. We look forward to publishing your paper in PLOS Biology. 

Sincerely, 

Roli

Roland G Roberts, PhD 

Senior Editor 

PLOS Biology